# Reasoning-as-Logic-Units: Scaling Test-Time Reasoning in Large Language Models Through Logic Unit Alignment

Cheryl Li [1]  Tianyuan Xu [2]  Yiwen Guo [1]

## Abstract

Chain-of-Thought (CoT) prompting has shown promise in enhancing the reasoning capabilities of large language models (LLMs) by generating natural language (NL) rationales that lead to the final answer. However, it struggles with numerical computation, which has somehow led to the development of program-aided techniques. Despite their potential, a persistent challenge remains: inconsistencies between LLM-reported reasoning steps and the logic in generated programs, which we term "reasoning hallucinations." This stems from the inherent ambiguities of NL and the statistical nature of LLMs, which often lack rigorous logical coherence. To address this challenge, we propose a novel test-time scaling framework, **Reasoning-as-Logic-Units (RaLU)**, which constructs a more reliable reasoning path by aligning logical units between the generated program and their corresponding NL descriptions. By decomposing the initially generated program into discrete units using static analysis, RaLU engages in an iterative dialogue with the LLM to judge, refine, and explain each unit. A rewind-and-correct mechanism ensures alignment between code statements and task requirements in each unit, ultimately forming a cohesive reasoning path under the program's logic, from which the model reaches a final solution. Our experiments demonstrate that RaLU significantly outperforms existing baselines in mathematical reasoning (GSM8K, MATH) and algorithmic reasoning (HumanEval+, MBPP+), underscoring its potential to advance LLM reasoning and programming by offering enhanced accuracy and interpretability.

[1]Independent Researcher, Beijing, China [2]Peking University, Beijing, China. Correspondence to: Yiwen Guo <guoyiwen89@gmail.com>.

*Proceedings of the 42nd International Conference on Machine Learning*, Vancouver, Canada. PMLR 267, 2025. Copyright 2025 by the author(s).

## 1. Introduction

Extensive studies have shown that Chain-of-Thought (CoT) (Wei et al., 2022) prompting can improve the reasoning capabilities of large language models (LLMs) (Mondorf & Plank, 2024; Fu et al., 2023; Liu et al., 2023a) by requiring LLMs to generate a rationale before its final decision. Complementary to CoT and its variants, program-aided techniques like Program-of-Thought (PoT) (Chen et al., 2023b) have emerged, which decompose complex reasoning and numerical computation by prompting LLMs to generate programs and use external interpreters to solve mathematical problems. When combined with test-time scaling, which dynamically allocates more computational resources during inference, these methods optimize LLM performance in reasoning tasks, particularly in mathematical and algorithmic domains (Zhong et al., 2024).

Despite these advancements, both CoT and PoT face significant challenges. The inherent ambiguity and imprecision in natural language (NL) impede precise calculations in CoT-like methods (Yu et al., 2024; Gao et al., 2023). Meanwhile, PoTs simply replace NL reasoning with programs, so they cannot improve LLM in code generation, yet solving algorithmic problems is an important aspect of LLM reasoning. Plus, crafting accurate programs in a single attempt remains challenging (Chen et al., 2024a), so PoT can even introduce more errors than CoT sometimes (Li et al., 2024).

Furthermore, simple combinations of CoT and PoT cannot yield satisfactory outcomes. Research indicates that guiding an LLM to generate step-by-step analysis in NL before deriving programs may not outperform direct prompting (Huang et al., 2024a). This underperformance can be traced to inconsistencies between reasoning steps and the logic in generated programs, which we term **"Reasoning Hallucinations."** The hallucinations manifest as: 1) accurate NL step descriptions but logical errors in individual code statements; 2) missing key steps or inclusion of irrelevant ones; and 3) correct steps misordered or improperly connected. Examples of these three types are presented in Figure 1.

These reasoning hallucinations arise from the statistical nature of LLMs, which generate responses based on token predictions rather than true reasoning. LLMs mimic reason-

**Type 1: Accurate NL step but logical errors in the code/mathematical expression**

**Question (MATH-np 794):** Find the smallest possible value of $\frac{(y-x)^2}{(y-z)(z-x)} + \frac{(z-y)^2}{(z-x)(x-y)} + \frac{(x-z)^2}{(x-y)(y-z)}$, where $x, y,$ and $z$ are distinct real numbers.

**Response (COT, DeepseekV3):**
### Step 1: Simplify the expression
Let's denote the expression as $E$.

We can rewrite each term by **factoring out** $-1$ from the denominators where necessary.

The first term becomes:
$\frac{(y-x)^2}{(y-z)(z-x)} = \frac{(y-x)^2}{-(z-y)(z-x)} = -\frac{(y-x)^2}{(z-y)(z-x)}$.

- The second term becomes
$\frac{(z-y)^2}{(z-x)(x-y)} = -\frac{(z-y)^2}{(z-x)(y-x)}$.

- The third term becomes:
$\frac{(x-z)^2}{(x-y)(y-z)} = -\frac{(x-z)^2}{(x-y)(y-z)}$. 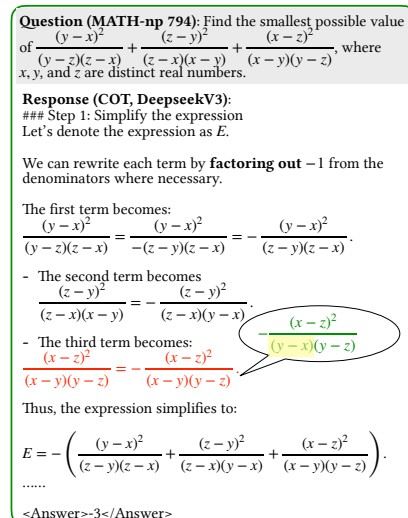 $\frac{(x-z)^2}{(y-x)(y-z)}$

Thus, the expression simplifies to:

$E = -\left( \frac{(y-x)^2}{(z-y)(z-x)} + \frac{(z-y)^2}{(z-x)(y-x)} + \frac{(x-z)^2}{(x-y)(y-z)} \right)$.
......

<Answer>-3</Answer>

**Error Analysis:** In Step 1, the AI correctly pinpoint to factor out $-1$ from the denominators to rewrite the expression as $E$. However, it mishandled the negative signs for the third term, i.e, the positions of $y$ and $x$ should be exchanged. This led to an erroneous simplification where the negative sign was applied twice, distorting the result. This error propagates to the final answer, which should be 3 instead.

---

**Type 2: Missing key steps in NL descriptions or code/mathematical expressions**

**Question (GSM8K 38):** John plans to sell all his toys and use the money to buy video games. He has 13 lego sets and he sells them for $15 each. He ends up buying 8 video games for $20 each and has $5 left. How many lego sets does he still have?

**Response (PoT, Llama-3.3):**
1. First, determine the total income from selling Lego sets lego sets. John has 13 Lego sets and he sells each one for $15. We need to know how much money he earns in total.
2. Calculate the total cost of video games: He buys 8 video games for $15 each. We need to find out how much money he spends on buying these video games.
3. Find the total money from selling Lego sets: if we add the money he spent on video games and the money he has left, we can get the total money he got from selling the Lego sets.
4. Calculate the remaining Lego sets: By subtracting the number of Lego sets he sold from the original number, we can find out how many Lego sets he still has.

```

total_legos = 13
lego_price = 15
video_game_price = 20
money_left = 5
# Calculate the total money John gets from selling the lego sets
total_money_earn = 13*15
# Calculate the total money he spends on video games
total_money_spent = 8 * video_game_price
remaining_money = total_money_earn - total_money_spent
# Since John sold all his lego sets, he should have 0 lego sets left.
ans = 0

```

`total_money_earn = total_money_spent + money_left`

**Error Analysis:** The critical missing step was solving for the actual number of lego sets sold based on the $5 leftover. In step 3, the description correctly indicates that calculating the earned money should use the leftover (added by the money spent). However, when writing the code, it assumed John sold all lego sets probably the problem states he "plans to sell all," without actually calculating the number of lego sets sold, leading to a wrong final answer.

---

**Type 3: Individually correct NL steps and correct code/expression, but wrong connections**

**Question (LeetCode 3200):** You are given two integers red and blue representing the count of red and blue colored balls. You have to arrange these balls to form a triangle such that the 1st row will have 1 ball, the 2nd row will have 2 balls, the 3rd row will have 3 balls, and so on. All the balls in a particular row should be the same color, and adjacent rows should have different colors. Return the maximum height of the triangle that can be achieved.

**Response (CoT, Llama-3.3, truncated):**
1. Try two starting color combinations: start with red or blue balls, handle them separately.
2. Build triangle in a loop: use starting-color balls for odd-numbered rows, different-color balls for even-numbered rows. If a row is built, increment row number and try the next.
3. Loop termination: subtract row - corresponding ball number from total. If remaining < 0, can't build the row, get max buildable rows by subtracting 1 from current row number.
4. After handling both combos, return the max triangle height achieved.

```

def maxHeightOfTriangle(self, red: int, blue: int) -> int:
    ans = 0
    for (x, y) in [(red, blue), (blue, red)]:
        i = 1
        while True:
            if i % 2 == 1: x -= i
            else: y -= i
            if x < 0: ans = max(ans, i-1); break
            if y < 0: ans = max(ans, i-1); break
            i += 1
    return ans

```

```
if i % 2 == 1:
    x -= i
    if x < 0:
        ans = max(ans, i-1)
        break
else:
    y -= i
    if y < 0:
        ans = max(ans, i-1)
        break
```

**Error Analysis:** The main issue with this code lies in the logic of checking the remaining number of balls. Specifically, the checks for whether x or y is less than 0 are placed after both the odd-row and even-row ball subtractions. This means that after a subtraction for an odd-numbered row makes x negative, the code will still proceed to check y for negativity, and vice versa. This can lead to incorrect results as the loop should terminate immediately once either color runs out of sufficient balls.

---

*Figure 1.* Motivating examples reflecting the reasoning hallucinations: In Example 1, the LLM attempts to factor out the negative sign as stated yet wrongly neglects to reverse the positions of $x$ and $y$ in the expression. In Example 2, the LLM overlooks the NL step that indicates using the leftover to calculate the money earned; instead, it is misled by the question description to sell all the sets, thus omitting this crucial step. In Example 3, the LLM incorrectly places the loop termination conditions after operating on both odd- and even-numbered rows, which should be positioned individually.

ing by reproducing patterns linked to logical explanations but are trained on text that often lacks rigorous logical coherence (Lenat & Marcus, 2023). The scarcity of detailed mapping from NL reasoning to precise, logical expressions like code or equations means LLMs struggle to capture the underlying logic patterns under former expressions, not to mention reproducing such patterns, resulting in such inconsistencies. Moreover, previous findings suggest this misalignment also affects other reasoning tasks, raising doubts about the authenticity of reported reasoning steps (Li et al., 2024). Unlike factual hallucinations that can be mitigated by introducing external information, reasoning hallucinations are intrinsic to the model's internal processing and pose a unique challenge in reliable LLM reasoning.

To overcome this challenge, we propose a novel reasoning framework that leverages programs as the logical skeletons and natural language as explanatory content. Our key insight is that if the two representations of reasoning processes, i.e., NL reasoning steps and generated programs, are aligned in the same fundamental logic, the reasoning path would be more reliable. Each reasoning step can be projected to a series of code statements, and the latter serves as the formalized implementation of the former. Hence, we introduce our test-time scaling framework, **Reasoning-as-Logic-Units (RaLU)**. Figure 2 compares its reasoning path to that of CoT, Self-consistency (Wang et al., 2023), and

Tree-of-Thought (ToT) (Yao et al., 2023) for illustration.

Specifically, our framework involves four core actions: self-reason, self-judge, self-explain, and self-correct, organized into three primary stages: **1) Logic Unit Extraction:** The framework begins by directly generating a program to address the given problem. This program serves as a representation of the reasoning process. Using a static analysis tool, we create a control flow graph (CFG) to depict the program's logic. RaLU traverses this CFG, dividing it into logical units based on program branches like conditional and looping statements. Each unit comprises several code statements, implementing operations for problem-solving. **2) Logic Unit Alignment:** RaLU initiates an iterative dialogue with the same LLM to assess the correctness of each logic unit. Beyond being a judge, the LLM explains the operations within each unit to ensure alignment with the problem specification. Should errors arise, the LLM self-corrects the unit, and the dialogue will rewind to the previous round for re-evaluation on the corrected unit. The reasoning path branches out until correctness is achieved or a predefined threshold is reached. **3) Solution Synthesis:** After processing all logic units, we obtain a reasoning path where each node is self-verified or contains a self-corrected version of code statements and NL explanations. Using this hybrid reasoning path as a conversation history, the LLM generates the final solution to the reasoning task.

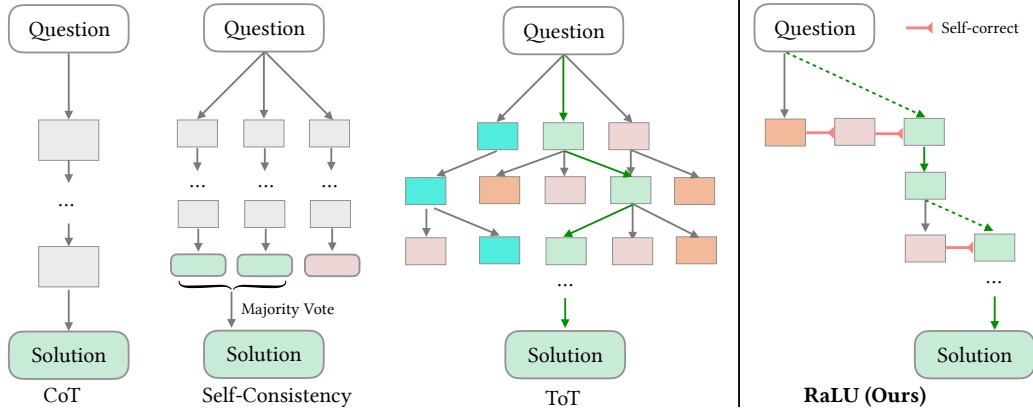

*Figure 2.* Schematic depicting multiple strategies for test-time scaling frameworks with LLMs. Each rectangular shape symbolizes a distinct thought (*aka.* step/unit), a self-contained text sequence crucial as an intermediate stage in the reasoning process. In previous studies, all the thoughts are natural language-based, while our RaLU uses logic units consisting of code statements and NL descriptions.

We evaluate RaLU on four benchmarks, including two for mathematical reasoning (GSM8K (Cobbe et al., 2021b), MATH (Hendrycks et al., 2021)) and the other two for code reasoning: HumanEval (Chen et al., 2021), MbPP (Austin et al., 2021), and their plus versions (Zhong et al., 2024). The evaluation involves three LLM backbones: Deepseek-V3, Llama3.3-70B-Instruct and Qwen2.5-72B-Instruct. Experimental results show that RaLU achieves a significant improvement in final answer accuracies or pass@1 compared with best-performing baselines, with specific improvement of 1.22%, 2.07%, 6.60%, and 2.17% on these four benchmarks, respectively. It is worth noting that RaLU outperforms the best-performing LLM family of reasoning models, o1, on HumanEval+ and MbPP+. We further perform an extensive ablation study to demonstrate the contributions to our key design in RaLU. Our code is available at https://github.com/DeepAccept/RaLU.

## 2. Related Works and Discussions

### 2.1. General Reasoning with LLMs

Prompting techniques have greatly improved the reasoning abilities of LLMs. CoT (Wei et al., 2022) is the most popular paradigm, deriving a large number of variants such as Least-to-Most (Zhou et al., 2023) and Auto-CoT (Zhang et al., 2023). The central concept of these approaches is "divide and conquer"–prompting LLMs to deconstruct complex problems into simpler sub-tasks, systematically address each one by reporting the process and then synthesize a comprehensive final answer. Some studies directly let LLMs write programs to serve as reasoning steps, such as PoT (Chen et al., 2023b) and Program-aided Language models (Gao et al., 2023), decoupling computation from reasoning and language understanding. However, they cannot improve the performance of LLMs in coding tasks and struggle with writing perfect programs within a single query, thus introducing more errors sometimes (Li et al., 2024). Ex-

isting studies have shown that simply mixing code and text during pre-training or instruction-tuning stages can enhance LLM reasoning (Ma et al., 2024), but how to effectively combine them remains under explosion.

### 2.2. Code Reasoning with LLMs

Inference-side approaches for coding tasks usually focus on debugging and refining the generated code since it is prone to logic errors, dead loops, and other unexpected behaviors. Many studies (Chen et al., 2023a; 2024b) generate unit tests or feedback from the same LLM to score and refine the generated programs, and ChatRepair (Xia & Zhang, 2023) relies on hand-writing test cases. Another stream of studies combines traditional software engineering tools to improve code quality, including executors (Zheng et al., 2024; Ni et al., 2023) and repair tools (Fan et al., 2023). Recent studies on multi-agent frameworks (Lee et al., 2024; Hong et al., 2024) also achieve advanced performance on coding tasks. They borrow the information provided by software analysis tools and embed such information into prompts to expand the ability bounds of LLMs in code reasoning.

### 2.3. Test-Time Scaling for LLM Reasoning

Recent studies have revealed that using more test-time computation can enable LLMs to improve their outputs (Snell et al., 2024). A primary mechanism is to select or vote the best CoT path from multiple independent sampling, such as Best-of-N sampling (Cobbe et al., 2021a) and Self-Consistency (Wang et al., 2023). Innovations like ToT (Yao et al., 2023), Graph-of-Thought (GoT) (Besta et al., 2024), and DeAR (Xue et al., 2024) design search-based schemes to expanding the range and depth of path exploration, though they are often suitable for specific tasks (e.g., the Game of 24) as they require to pre-define a fixed candidate size for each node, leading to redundancy or insufficiency.

Another stream of research scales inference time by en-

abling models to critique and revise their answers iteratively, which has been applied in general reasoning tasks (Kamoi et al., 2024; Liu et al., 2024). Intrinsic self-correction asks LLMs to identify and fix errors based on their inner knowledge without any external tools or information, such as Self-Check (Miao et al., 2024), Self-Refine (Ranaldi & Freitas, 2024), and StepCo (Wu et al., 2024). External self-correction allows for tool usage such as code interpreters and search engines (Gou et al., 2024; Ding et al., 2024). Recent studies have reported that intrinsic self-correction may struggle with judging or modifying their own responses (Kamoi et al., 2024; Huang et al., 2024b). Yet, a more recent empirical study shows that intrinsic self-correction capabilities are exhibited across multiple existing LLMs under fair prompting–do not directly or indirectly influence the LLM to change or maintain its initial answer (Liu et al., 2024).

# 3. Reasoning-as-Logic-Units

We propose a novel structured test-time scaling framework, RaLU, which enforces alignment between NL descriptions and code logic to leverage both sides. Programs ensure rigorous logical consistency through syntax and execution constraints, whereas NL provides intuitive representations with problem semantics and human reasoning patterns.

Specifically, RaLU operationalizes this synergy through three iterative stages (as shown in Figure 3): *Logic Unit Extraction*, *Logic Unit Alignment*, and *Solution Synthesis*. The first stage decomposes an initially generated program into atomic logic units via static code analysis. Then, an iterative multi-turn dialogue engages the LLM to 1) explain each unit's purpose in NL, grounding code operations in problem semantics, 2) validate computational correctness and semantic alignment with task requirements, and 3) correct errors via a rollback-and-revise protocol, where detected inconsistencies trigger localized unit refinement. The validated units form a cohesive, executable reasoning path. The final stage synthesizes this path into a human-readable solution, ensuring the final answer inherits the program's logical rigor while retaining natural language fluency.

In this way, RaLU can significantly mitigate reasoning hallucinations. First, each unit seamlessly pairs executable code with NL explanations to address the type-one hallucination through explicitly aligning local logic. Second, the LLM focuses on only one unit per response in case of missing a crucial step or introducing an irrelevant step, and iterative verification ensures that the LLM notices all problem constraints. Third, these logic units are interconnected rigorously along the program structure, ensuring logical coherence of the reasoning path.

In summary, by structurally enforcing bidirectional alignment between code logic and textual justifications, we build a self-consistent reasoning path where computational validity and conceptual clarity mutually reinforce each other. This architecture not only minimizes logical discrepancies but also provides transparent intermediate steps for error diagnosis and refinement.

## 3.1. Logic Unit Extraction

RaLU begins with prompting the LLM to generate an initial program that serves as a reasoning scaffold for the task. While possibly imperfect, this program approximates the logical flow required to derive a solution, providing a structured basis for refinement.

We apply static code analysis to construct a Control Flow Graph (CFG), where nodes represent basic blocks (sequential code statements), and edges denote control flow transitions (e.g., branches, loops). A CFG explicitly surfaces a program's decision points and iterative structures, whose details are illustrated in Appendix A.2. RaLU then partitions the code into atomic units by dissecting the CFG at critical junctions—conditional blocks (if/else), loop boundaries (for/while), and function entries. Each unit encapsulates a self-contained computational intent, such as iterating through a list or evaluating a constraint.

## 3.2. Logic Unit Alignment

The alignment stage iteratively validates and refines logic units through a stateful dialogue governed by:

$$\mathcal{V}_i = \text{LLM}\Big( \underbrace{\mathcal{S}}_{} \oplus \underbrace{\bigoplus_{k=0}^{i-1} \mathcal{U}_k}_{} \oplus \underbrace{\mathcal{P}(\mathcal{U}_i)}_{} \Big) \qquad (1)$$

where $\mathcal{U}_i$ denotes the $i$-th unit, $\mathcal{S}$ is the task specification, and the operator $\oplus$ represents contextual concatenation. $\mathcal{P}(\mathcal{U}_i)$ instructs the LLM to handle the $i$-th unit, where each turn of interaction is responsible for judging the correctness, modifying it upon errors, and explaining it to align with the task specification. Thus, each response $\mathcal{V}_i = \langle \mathcal{J}_i, \tilde{\mathcal{U}}_i \rangle$ comprises a judgment token $\mathcal{J}_i \in \{\text{OK}, \text{WRONG}\}$ and a refined unit $\tilde{\mathcal{U}}_i$. The refinement adheres to:

$$\tilde{\mathcal{U}}_i = \begin{cases} \mathcal{U}i & \text{if } J_i = \text{OK} \\ \text{LLM}_{\text{repair}}(\mathcal{S}, \mathcal{U}_i, \tilde{\mathcal{U}}_k, \ k < i) & \text{otherwise} \end{cases} \qquad (2)$$

To prevent error cascades, corrections trigger a partial rewind: the original unit $\mathcal{U}_i$ is replaced by the refined version $\tilde{\mathcal{U}}_i$ in the interested reasoning path. Then, $\tilde{\mathcal{U}}_i$ will be re-validated based on previous units $\{\mathcal{U}_k | k < i\}$. This aims to construct a path $\mathcal{P}$ with all nodes able to pass self-judging:

$$\forall \mathcal{U}_k \in \mathcal{P} = \{\mathcal{U}_1, \cdots, \mathcal{U}_{i-1}\}, \quad \mathcal{J}_k = \text{OK}. \qquad (3)$$

The correctness process terminates under two conditions: 1) fixed-point convergence, i.e., all units satisfy $J_i =$

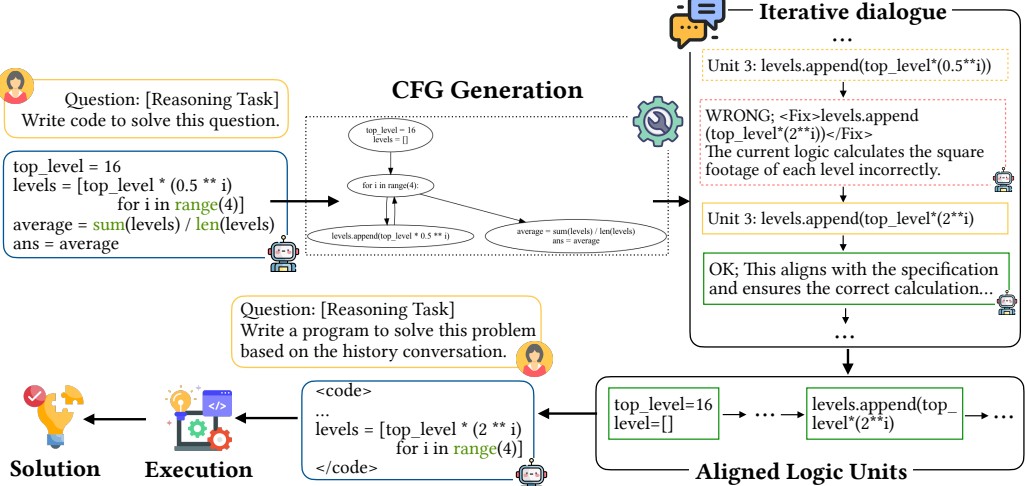

*Figure 3.* Illustrating the three-stage process of RaLU: Logic Unit Extraction, Logic Unit Alignment, and Solution Synthesis for operationalizing synergy in reasoning tasks.

$\text{OK} \wedge \tilde{\mathcal{U}}_i = \mathcal{U}_i$, indicating that no further are refinements needed; and 2) a predefined iteration limit or confidence threshold is reached. Upon triggering the second condition, multiple candidate units will exist, and we select the optimal version $\tilde{\mathcal{U}}_i^*$ using a normalized confidence metric. In this case, there are multiple candidates for a unit, and none of them has been judged as correct. We select the most confident response. The confidence score is calculated as the following equation 4, based on the log probabilities, which express token likelihoods on a logarithmic scale $(-\infty, 0]$, reported by the LLM.

$$\text{Conf}(\tilde{\mathcal{U}}) = \frac{1}{n} \sum_{j=1}^{n} \sigma(lp_j),$$

$$\sigma(lp_j) = \min\left(e^{lp_j} + 0.005, 1\right) \times 10^{-2}. \tag{4}$$

where $lp_j$ denotes the log probability of the $j$-th token in the LLM's response, mapped to a [0,1] scale via the clamping function $\sigma$. For LLMs lacking log probability outputs, we employ a self-consistency checking process–prompting the same LLM ranks candidates to determine $\tilde{\mathcal{U}}_i^*$.

Herein, we discuss whether $\tilde{\mathcal{U}}$ is more likely to be correct than its original version $\mathcal{U}$ for any unit, that is $P(\mathcal{U} \text{ is correct}) = p < P(\tilde{\mathcal{U}} \text{ is correct}) = p'$. Let's define $\alpha = P(J(\mathcal{U}) = \text{OK}|\mathcal{U} \text{ is correct})$ (true positive rate) and $\beta = P(J(\mathcal{U}) = \text{WRONG}|\mathcal{U} \text{ is incorrect})$ (true negative rate). Thus, we have:

$$p' = \alpha p + \gamma_{repair}[(1 - \alpha)p + \beta(1 - p)] \tag{5}$$

where $\gamma_{repair} = P(R(\mathcal{U}) \text{ is correct}|J(U) = \text{WRONG})$ with $R(\cdot)$ representing the LLM's repair action. Then, the condition of $p' > p$ is transformed as:

$$\gamma_{repair} > P(\mathcal{U} \text{ is correct}|J = \text{WRONG}). \tag{6}$$

See Appendix A.3 for the detailed derivation. Empirical studies show that modern LLMs can achieve high accuracies when serving as a judge (Thakur et al., 2024) (where $\alpha$ can reach 0.9+), so the above condition can be easily achieved with intelligent LLMs. Nevertheless, if the model is almost perfect ($p \approx 1$), then using RaLU cannot make significant improvement even though ($p' > p$).

In addition to evaluating and refining the unit, the LLM is tasked with generating explanations that explicitly map the unit's behavior to the task specification. These explanations serve two critical roles. First, they help to justify whether the unit aligns with or violates the intended logic. Second, they demystify the reasoning process, exposing the LLM's thinking about execution behavior in human-interpretable terms. By linking concrete code elements to abstract specification requirements, the LLM acts as a translator between implementation and intent. This dual focus on correctness and explainability ensures that both the code and its rationale evolve cohesively during refinement.

### 3.3. Solution Synthesis

Through logic unit alignment, RaLU constructs a coherent sequence of self-verified operations paired with precise NL explanations. This establishes a unified reasoning path that integrates computational logic with interpretive alignment (with problem specifications), ensuring rigorous consistency between code behavior and reasoning steps. Guided by this aligned reasoning path, the LLM synthesizes the structured units into a final solution using the following prompt: *"Based on the previously self-verified reasoning path, generate a correct program to solve the given problem."*

This dual-anchoring mechanism–enforcing program-executable logic and specification-aligned reasoning– eliminates ambiguities for response generation. We formalize the effectiveness of RaLU through a Bayesian inference lens, demonstrating how iterative logic unit

alignment systematically amplifies the likelihood of generating correct programs.

Let $C$ denote the event where the LLM produces a program correctly solving the task, and $\overline{C}$ its complement. Each logic unit $O_i(1 \leq i \leq n)$ represents a self-verified reasoning step aligned with both program execution and problem semantics. By Bayes' theorem, the posterior probability of correctness, conditioned on validated units, is:

$$P(C|O_1, \ldots, O_n) = \frac{P(O_1, \ldots, O_n|C) \cdot P(C)}{P(O_1, \ldots, O_n)}$$
$$= \frac{P(O_1, \ldots, O_n|C) \cdot P(C)}{P(O_1, \ldots, O_n|C)P(C) + P(O_1, \ldots, O_n|\overline{C})P(\overline{C})}$$
(7)

Note that a correct program inherently exhibits logical coherence, making its reasoning steps more likely to align with human-judged validity, so $P(O_1, \cdots, O_n|C) \gg P(O_1, \cdots, O_n|\overline{C})$. This asymmetry implies:

$$\frac{P(O_1, \ldots, O_n|C)}{P(O_1, \ldots, O_n)} \geq 1 \implies P(C|O_1, \ldots, O_n) > P(C)$$
(8)

Hence, RaLU 's rewind-and-correct mechanism—by enforcing consistency across units—statistically elevates the prior correctness probability $P(C)$ (initial program quality) to a higher posterior $P(C|O_1, \cdots, O_n)$. This Bayesian progression quantifies how structured, self-validated reasoning suppresses hallucinations, ensuring solutions inherit rigor from aligned logic units.

Crucially, even if generating incorrect solutions, RaLU maintains granular traceability through self-contained logic units. This enables precise identification of defective components responsible for errors, rooted in the framework's transparency. By transforming black-box reasoning into more debuggable processes, RaLU accelerates error correction and enhances interpretability for human-AI collaboration.

# 4. Experiments

## 4.1. Experiment Setup

**Benchmarks.** We use five benchmarks: three for mathematical reasoning: GSM8K (Cobbe et al., 2021b), MATH (Hendrycks et al., 2021), and AQUA (Ling et al., 2017), as well as the other two for code reasoning, HumanEval (Chen et al., 2021) and Mbpp (Austin et al., 2021), along with their extended versions with more test cases (Liu et al., 2023b). See Appendix B.1 for more details about the benchmarks. We evaluate RaLU on the whole test set except MATH. Due to resource limitation, we follow (Miao et al., 2024) to use a subset of MATH (named by MATH-np) taken

from (Ling et al., 2023) [1]. We use the metrics of the answer accuracy and pass@1 score for math- and code-reasoning, respectively. All the experiments are conducted three times independently. We report the average result in the format of $\mu(\pm\sigma)$, where $\mu$ is the mean, and $\sigma$ is the standard deviation. Our focus on math- and code-reasoning is due to the availability of well-established benchmarks and the ease of evaluating outputs. RaLU can be directly applicable to other domains with minimal adjustments to the prompts.

**Baselines.** We compare RaLU against three categories of baselines without fine-tuning or external information: 1) promoting methods for general purposes: Direct Prompting, Zero-Shot CoT (Wei et al., 2022), ToT (Yao et al., 2023), and Self-Consistency (SC) (Wang et al., 2023). 2) self-correction-based approaches: Self-Calibration (Scal) (Kadavath et al., 2022), and Self-Refine (SR) (Ranaldi & Freitas, 2024); 3) techniques specific for either task: PoT (Chen et al., 2023b), Self-Check (SCheck) (Miao et al., 2024), and rubber-duck debugging derived from Self-Debug (SD) (Chen et al., 2024b). Details of these baselines are provided in Appendix B.2.

**Implementation.** We deploy RaLU on three open-source LLMs: Deepseek-V3 (Dec 2024), Qwen2.5-72B-Instruct (Sep 2024), and Llama3.3-70B-Instruct (April 2024). To prevent breaches of anonymity, we do not deploy RaLU on commercial closed-source models such as GPTs and o1. Instead, we compare RaLU with the public results of these closed-source LLMs reported on the leaderboard maintained by (Liu et al., 2023b; Paperwithcode, 2025; Mirzadeh et al., 2024), presented in Appendix C. We set the maximum number for self-correction turns as 3 and the maximum number of candidate solutions/branches as 10 for Self-Consistency and ToT. The temperature parameter is set to 0.7, and the frequency penalty is 0.3 in all experiments.

## 4.2. Results

Table 1 summarizes the performance of RaLU on math and code reasoning. Across all benchmarks and diverse LLM architectures, RaLU consistently outperforms existing baselines, demonstrating its generalizability and robustness. We analyze the advantages through three critical comparisons:

**RaLU v.s. Single-Path Reasoning.** Compared to direct prompting, CoT, and PoT (single reasoning path per query), RaLU achieves an average improvement of +12.81% and +14.85% for math and code reasoning, respectively, attributed to its structured decomposition of problems into logical units aligned with programmatic constraints, mitigating the inconsistencies inherent in linear reasoning chains,

---

[1] `https://github.com/lz1oceani/verify_cot/blob/main/results/chatgpt3.5/natural_program/MATH_np.json`

*Table 1.* RaLU significantly increases final scores with all the three LLM backbones. ΔGain is the performance gain of RaLU compared with the best-performing baseline.

| Dataset | LLM | Direct | CoT | ToT | PoT/SR⋆ | SC | SCal | SCheck/SD⋆ | RaLU | ΔGain(%) |
|---------|-----|--------|-----|-----|---------|-----|------|------------|------|----------|
| GSM8K | DeepSeek V3 | 0.917 | 0.968 | 0.921 | 0.939 | 0.965 | 0.949 | 0.942 | **0.971** | +0.310 |
| | Qwen2.5-72B | 0.940 | 0.967 | 0.937 | 0.951 | 0.964 | 0.945 | 0.901 | **0.980** | +1.344 |
| | Llama3.3-72B | 0.826 | 0.944 | 0.904 | 0.876 | 0.945 | 0.928 | 0.881 | **0.964** | +2.011 |
| MATH-np | DeepSeek V3 | 0.691 | 0.723 | 0.643 | 0.751 | 0.703 | 0.814 | 0.601 | **0.821** | +0.860 |
| | Qwen2.5-72B | 0.706 | 0.708 | 0.689 | 0.753 | 0.780 | 0.726 | 0.693 | **0.791** | +1.410 |
| | Llama3.3-72B | 0.347 | 0.541 | 0.489 | 0.551 | 0.607 | 0.539 | 0.533 | **0.631** | +3.954 |
| AQUA | DeepSeek V3 | 0.768 | 0.811 | 0.772 | 0.799 | 0.839 | 0.807 | 0.783 | **0.843** | +0.420 |
| | Qwen2.5-72B | 0.764 | 0.799 | 0.791 | 0.807 | 0.779 | 0.811 | 0.772 | **0.846** | +4.316 |
| | Llama3.3-72B | 0.512 | 0.583 | 0.555 | 0.496 | **0.610** | 0.598 | 0.528 | **0.610** | 0 |
| HumanEval | DeepSeek V3 | 0.915 | 0.915 | 0.890 | 0.854 | 0.933 | 0.921 | 0.878 | **0.939** | +0.643 |
| | Qwen2.5-72B | 0.841 | 0.872 | 0.841 | 0.707 | 0.738 | 0.787 | 0.774 | **0.909** | +4.243 |
| | Llama3.3-72B | 0.689 | 0.713 | 0.704 | 0.591 | 0.713 | 0.701 | 0.585 | **0.811** | +13.745 |
| HumanEval+ | DeepSeek V3 | 0.878 | 0.878 | 0.835 | 0.805 | 0.884 | 0.866 | 0.799 | **0.902** | +2.036 |
| | Qwen2.5-72B | 0.793 | 0.793 | 0.823 | 0.659 | 0.707 | 0.750 | 0.738 | **0.860** | +4.496 |
| | Llama3.3-72B | 0.628 | 0.652 | 0.653 | 0.512 | 0.671 | 0.646 | 0.543 | **0.768** | +14.456 |
| Mbpp | DeepSeek V3 | 0.923 | 0.926 | 0.910 | 0.892 | 0.921 | 0.915 | 0.899 | **0.937** | +1.188 |
| | Qwen2.5-72B | 0.921 | 0.894 | 0.910 | 0.862 | 0.921 | 0.918 | 0.902 | **0.926** | + 0.543 |
| | Llama3.3-72B | 0.807 | 0.828 | 0.782 | 0.738 | 0.823 | 0.831 | 0.775 | **0.836** | +0.598 |
| Mbpp+ | DeepSeek V3 | 0.783 | 0.778 | 0.759 | 0.720 | 0.791 | 0.786 | 0.767 | **0.833** | +5.310 |
| | Qwen2.5-72B | 0.791 | 0.765 | 0.778 | 0.730 | 0.775 | 0.780 | 0.791 | **0.828** | +4.678 |
| | Llama3.3-72B | 0.664 | 0.704 | 0.601 | 0.550 | 0.698 | 0.685 | 0.638 | **0.709** | +0.710 |

⋆ Using the former method for math reasoning and using the latter method for code generation.

either represented in NL (CoT) or programs (PoT). We display more cases about how RaLU reduces reasoning hallucinations of the combination of CoT and PoT in Appendix D.1. RaLU enables fuller exploration in the diverse solution subspaces, resulting in optimal solution generation.

**RaLU v.s. Multi-Path Exploration.** Multi-path methods like Self-Consistency, Self-Check, and ToT aim to select the optimal reasoning path over multiple samples. SC and ToT rely on sampling fixed times of independent candidates (up to 10 paths/branches), yet RaLU surpasses them by +9.55% and +10.69% for math and code reasoning, respectively, with far fewer candidates ($\leq 3$ per unit). This is because RaLU reduces cascading errors by isolating and refining individual units with hybrid reasoning representations. In contrast, SC or ToT might aggregate multiple incorrect paths that share the same flawed premise.

While Self-Check improves robustness through weighted voting—prioritizing solutions with internally consistent steps—it suffers from two critical limitations: First, its step-wise regeneration and comparison decorrelate errors but fail to propagate corrected logic to subsequent steps. Second, each re-generation requires 3+ LLM calls with redundant contexts, incurring high costs without guaranteeing holistic consistency. RaLU addresses these via unit-level iterative refinement. By decomposing reasoning into logical units,

errors are localized and resolved before subsequent units are processed while reducing LLM calls by about 60+%. A refined unit $i$ directly informs the context for unit $i + 1$, preventing error propagation. This enables RaLU to outperform Self-Check by 15.07% on average, achieving accuracy and efficiency through context-aware, incremental validation.

**RaLU v.s. Self-Correction Methods.** Many existing self-correction-based methods (e.g., Self-Refine and Self-Debug), often degrade performance by introducing errors into initially correct responses–a flaw exacerbated by their assumption of imperfection existence in the initial response attempt. RaLU mitigates this via a self-judgment stage, where LLMs validate each unit before refinement. This proactive verification yields an average gain of 18.28% over these baselines. Though Self-Calibration applies holistic self-judgment to reduce wrong edits, it still underperforms RaLU by 6.13% and 9.11% for math and code, respectively, as end-to-end validation fails to isolate localized inconsistencies addressed by RaLU 's granular, unit-level verification.

In summary, RaLU outperforms baselines by combining formal correctness (via code) and interpretability (via NL). This dual approach enables precise, self-contained error correction, significantly mitigating reasoning hallucinations.

**Multiple runs.** Given the inherent stochastic nature of

*Table 2.* The repeated three runs of experiments using Qwen-72-Instruct on Mbpp/Mbpp+ and Math-np. In each cell in the format of $\mu \pm \sigma$, $\mu$ is the mean, and $\sigma$ is the standard deviation.

| Dataset | Direct | CoT | ToT | PoT/SR* | SC | SCal | SCheck/SD | RaLU |
|---|---|---|---|---|---|---|---|---|
| Mbpp | 0.923±0.0070 | 0.895±0.0014 | 0.905±0.0045 | 0.860±0.0046 | 0.922±0.0033 | 0.924±0.0038 | 0.905±0.0038 | **0.957**±0.0012 |
| Mbpp+ | 0.788±0.0021 | 0.761±0.0052 | 0.772±0.0046 | 0.725±0.0038 | 0.779±0.0061 | 0.787±0.0122 | 0.750±0.0349 | **0.856**±0.0017 |
| Math-np | 0.705±0.0065 | 0.701±0.0085 | 0.695±0.0045 | 0.743±0.023 | 0.764±0.014 | 0.725±0.0065 | 0.685±0.0088 | **0.803**±0.0085 |

*Table 3.* Average token consumption of RaLU and baselines on MATH-np using Qwen-72B-Instruct. In each cell, $i/o$ denotes the average input and output token consumption, respectively. RaLU consumes 15x tokens compared to CoT, while saves 10x tokens compared to multi-path reasoning baselines such as Self-Check and ToT.

| Task | Direct | CoT | ToT | PoT | SC | SCal | SCheck | RaLU |
|---|---|---|---|---|---|---|---|---|
| MATH-np | $8\times10^4/7\times10^4$ | $8\times10^4/5.5\times10^5$ | $7.5\times10^7/1.2\times10^7$ | $2\times10^5/9\times10^4$ | $1.3\times10^6/3\times10^6$ | $1.5\times10^5/8\times10^4$ | $2\times10^7/1.4\times10^6$ | $4.8\times10^6/7\times10^5$ |

LLMs and the absence of seed support in the official API, we conducted two extra independent runs to further validate the robustness of RaLU across different model configurations. To conserve computational resources, we performed additional trials on Qwen-72B-Instruct for the MATH and MBPP/MBPP+ benchmarks, rather than repeating the full set of experiments. The results—reported as mean ± standard deviation in Table 2—are consistent with our findings, further confirming RaLU's stable and reliable performance.

**Inference Cost.** This section compares the inference token consumption of RaLU and baselines on MATH-np using Qwen-72B-Instruct, as dataset MATH is complex enough to require the most token consumption. As shown in Table 3, RaLU consumes 15x tokens compared to CoT, while saving 10x tokens compared to multi-path reasoning baselines such as Self-Check and ToT.

## 5. Ablation Studies

### 5.1. CFG v.s. Line-by-line

To validate the influence of the granularity of the logic unit on RaLU, we replace the CFG-driven decomposition with a line-by-line approach, treating each code line in the originally generated program as an independent logic unit. As displayed in Figure 4, results show an average performance decline of 7.04% across all benchmarks on Llama3.3, alongside a 37.7% increase in token consumption.

The observed decline stems from three intrinsic limitations of line-by-line decomposition. First, programs inherently consist of interdependent code blocks (e.g., loops, conditionals). Splitting them into isolated lines disrupts contextual dependencies between statements. Second, while fine-grained units obscure the hierarchical structure of program logic, LLMs struggle to associate low-level symbol operations (e.g., variable updates) with high-level problem-solving goals (e.g., iterative summation), leading to fragmented explanations and misaligned corrections. Third, line-by-line units amplify error accumulation. For example, a variable initialization error in line 1 may invalidate subsequent lines.

However, independent unit verification delays error detection, requiring repetitive corrections across multiple units. In contrast, CFG-based grouping localizes errors within bounded logical scopes.

The surge in token usage is intuitive. Each line triggers a separate verification dialogue, multiplying interaction rounds. Moreover, the LLM repeatedly re-encounters overlapping contexts and generates similar NL descriptions across units, wasting tokens on redundant information.

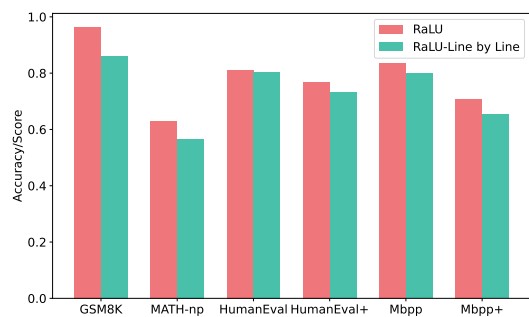

*Figure 4.* Ablation study of logic unit granularity: line-by-line decomposition causes 7.04% performance decline and 37.7% more token overhead compared to the CFG method (Llama). Performance degradation reflects contextual fragmentation and error propagation in atomic units, while increased token costs are attributed to redundant context re-verification.

### 5.2. NL Steps v.s. Logic Units

To further validate the necessity of program-guided logic units, we remove the initial program generation phase and instead treat each natural language reasoning step under the CoT prompting as an independent unit. This ablation leads to a 5.52% accuracy drop on mathematical tasks and 4.35% score drop on code reasoning, as shown in Figure 5, directly attributable to exacerbated reasoning hallucinations, The amplified decline highlights the fundamental limitations of pure natural language reasoning units.

We find a significant number of wrong answers can be attributed to reasoning hallucinations. First, NL steps like "compute the average by dividing the sum by the count"

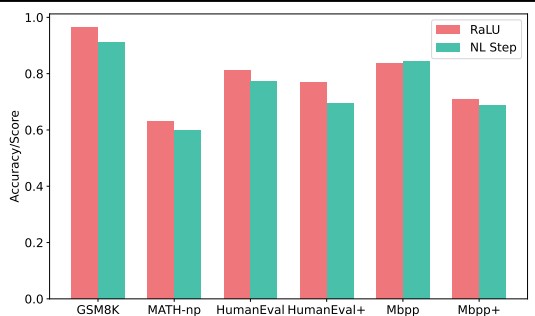

*Figure 5.* Ablation on unit abstraction: 5.52% accuracy drop (Math) and 4.35% score decline (Code) when replacing program-guided logic units with NL steps. Performance deterioration stems from reasoning hallucinations exacerbated by NL's lack of operational specificity and weak causal dependency constraints.

often lack operational specificity. While the NL step appears correct, the generated code may implement flawed logic (e.g., total / len(items) without handling empty lists). Unlike CFG units, which enforce alignment through static code analysis, the free-form language allows the LLM to hallucinate plausible-but-incorrect implementations. Moreover, the ambiguity of NL enables conceptual bundling–multiple logical operations (e.g., loop initialization, iteration, termination)–may be compressed into a single step like "iterate through the list." This can lead to code with missing boundary checks or redundant variables, as the LLM fails to decompose high-level descriptions into executable sub-operations. In addition, the NL narrative poorly constrains causal dependencies. For example, a step "update the total after checking a certain condition" might lead to code that evaluates the condition after modifying the total. CFG-driven units prevent such misordering by structurally embedding control flows. Appendix D.2 provides a detailed case study of how reasoning hallucinations are introduced if the program-driven logic units in RaLU are replaced by NL steps generated through CoT.

### 5.3. Candidate Unit Selection

In some complex questions, the number of self-corrections reaches the predefined threshold, necessitating the selection of a suboptimal reasoning path before all units are verified. In this paper, we use the confidence score (introduced in Section 3.2) as the selection criterion. We conducted ablation studies on Qwen-72B on MATH-np, which contains a sufficient number of challenging cases (89 out of 700) where the threshold is reached. We compared three alternative strategies: random selection, selecting the candidate with the lowest perplexity ($=: \exp(-1 * \text{mean of token probabilities})$), and selecting the last generated candidate.

As shown in Table 4, the impact of the selection strategy is minimal. Two factors can explain this. First, during self-correction iterations, LLMs tend to generate tokens with con-

sistently high probabilities (averaging above 0.9), leading to small variations between confidence scores and perplexity. Second, qualitative analysis reveals that many generated candidates are semantically equivalent, differing only in implementation details. This explains why even random selection results in only minor performance drops. Although perplexity-based selection slightly improves performance (+0.9%), the marginal gain suggests that the verification-revision loop already filters out most critical errors before selection takes place.

*Table 4.* Experiments with Qwen-72B-Instruct on the 89 complex cases from MATH-np, where RaLU reaches the budget limit of generating candidates. In each cell, $x/y$ indicates $x$ correct cases out of the 89 ones. The results show that the selection strategy has little impact on the final results.

| Confidence (RaLU) | Random | Perplexity | Last |
|---|---|---|---|
| 42/89=0.472 | 40/89=0.449 | 45/89=**0.506** | 38/89=0.427 |

## 6. Conclusion

We present Reasoning-as-Logic-Units (RaLU), a pioneering test-time scaling framework designed to tackle the issue of reasoning hallucinations and enhance the reasoning capabilities of LLMs. Unlike existing methods that often encounter logical inconsistencies between reported reasoning steps and generated programs, RaLU effectively extracts logic units from generated programs and aligns them with task requirements using natural language explanations. This method leverages the strengths of both natural language and program logic, resulting in more reliable, interpretable, and transparent LLM reasoning. Experimental results demonstrate that RaLU consistently outperforms existing baselines across various LLMs, including comparisons with proprietary close-sourced models. We hope that our work will inspire further research into structural reasoning, advancing LLM problem-solving across diverse domains.

## Impact Statement

This paper presents work whose goal is to advance the field of machine learning by improving the reliability and accuracy of large language models (LLMs) in complex reasoning tasks. By addressing reasoning hallucinations through logic-aligned hybrid reasoning processes, our framework enhances LLMs' general capabilities to generate coherent and logically consistent solutions, particularly in mathematical and algorithmic domains, without any fine-tuning or re-training. Potential societal benefits include more trustworthy AI systems for education, technical problem-solving, and decision-support applications. There are many broader societal consequences of our work, none of which we feel must be specifically highlighted here.

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

# A. Illustrations of RaLU

## A.1. A Complete Example

This section showcases how RaLU solves a code generation task using an LLM (e.g., deepseek V3 in this example). The question comes from Mbpp 77. In the initial response, RaLU simply uses direct prompting to ask the LLM to write a program for the given specification.

**Original Attempt.** [User] You are an expert in Python coding. Wrap your program in a   block. No more test cases or any other contents.

Specification: Write a function to find the Eulerian number a(n, m), the entry point is "eulerian_num".
assert eulerian_num(3, 1) == 4

[AI] 
```
1   def eulerian_num(n, m):
2       if m < 0 or m >= n:
3           return 0
4       if n == 0:
5           return 1
6       return (n - m) * eulerian_num(n - 1, m - 1) + (m + 1) * eulerian_num(n - 1, m)
```


Afterward, we extract Control Flow Graph (CFG) from the initial program, and then we transverse the CFG to transform it into text by describing the logic flow of the code statements. Therefore, we obtain a list of logic units:

- #ENTER FUNCTION# eulerian_num
  #BRANCH# If Condition 'if m < 0 or m >= n' is satisfied, then RUN 'return 0'

- #BRANCH# Otherwise, when Condition 'if m < 0 or m >= n' is not satisfied, then #BRANCH# If Condition 'if n == 0' is satisfied, then RUN 'return 1'

- #BRANCH# Otherwise, when Condition 'if n == 0' is not satisfied, then RUN 'return (n - m) * eulerian_num(n - 1, m - 1) + (m + 1) * eulerian_num(n - 1, m)'
  #EXIT FUNCTION#

Then, RaLU initializes a continuous dialogue with the same LLM. In each turn of interaction, the LLM is promoted to judge the correctness of the current logic unit. If this unit is determined as wrong, the LLM will generate the fixed version simultaneously.

**U1-I.** [User] ## Process
Unit 1: #ENTER FUNCTION# eulerian_num
#BRANCH# If Condition 'if m < 0 or m >= n' is satisfied, then RUN 'return 0'

**[AI] (confidence: 96.50)** OK
Analysis: This step correctly handles the edge case where 'm' is less than 0 or greater than or equal to 'n'. According to the specification, the Eulerian number 'a(n, m)' should return 0 in such cases. The logic aligns with the requirement, so this step is correct.

The correctness of unit 1 is checked by the LLM.

**U2-I.** [User] Unit 2: #BRANCH# Otherwise, when Condition 'if m < 0 or m >= n' is not satisfied, then #BRANCH# If Condition 'if n == 0' is satisfied, then RUN 'return 1'

**[AI] (confidence: 98.96)** WRONG
<Fix>
#BRANCH# Otherwise, when Condition 'if m < 0 or m >= n' is not satisfied, then #BRANCH# If Condition 'if m == 0' is satisfied, then RUN 'return 1'

</Fix>
Analysis: This unit incorrectly checks for the condition 'if n == 0'. According to the specification and the logic of Eulerian numbers, when 'm == 0', the function should return 1. The condition should be checking if 'm == 0', not 'n == 0'. This correction aligns with the requirement.

Herein, the reasoning path branches off by fixing the original logic unit. Then we rewind and replace unit 2 with the fixed one:

**U2-II.** [User] Unit 2: #BRANCH# Otherwise, when Condition 'if m ¡ 0 or m ¿= n' is not satisfied, then #BRANCH# If Condition 'if m == 0' is satisfied, then RUN 'return 1'

**[AI] (confidence: 97.61)** OK
Analysis: This unit correctly handles the case where 'm == 0'. According to the specification, when 'm == 0', the Eulerian number 'a(n, m)' should return 1. The logic aligns with the requirement, so this unit is correct.

**U3-I.** [User] Unit 3: #BRANCH# Otherwise, when Condition 'if n == 0' is not satisfied, then RUN 'return (n - m) * eulerian_num(n - 1, m - 1) + (m + 1) * eulerian_num(n - 1, m)' #EXIT FUNCTION#

**[AI] (confidence: 98.70)** WRONG
<Fix>
#BRANCH# Otherwise, when Condition 'if m == 0' is not satisfied, then RUN 'return (n - m) * eulerian_num(n - 1, m - 1) + (m + 1) * eulerian_num(n - 1, m)'
#EXIT FUNCTION#
</Fix>
Analysis: The unit incorrectly checks for 'if n == 0', which is already handled in the first branch ('if m < 0 or m >= n'). Instead, it should proceed to calculate the Eulerian number using the recursive formula when 'm != 0'. The corrected step ensures that the recursive calculation is performed only when 'm != 0', aligning with the specification.

In the dialogue round for unit 3, given that the previous historical conversation could serve as a priori conditions for the LLM, it successfully carried forward the modifications made to unit 1, thus ensuring the consistency of the context.

**U3-II.** [User] Unit 3: #BRANCH# Otherwise, when Condition 'if m == 0' is not satisfied, then RUN 'return (n - m) * eulerian_num(n - 1, m - 1) + (m + 1) * eulerian_num(n - 1, m)' #EXIT FUNCTION#

**[AI] (confidence: 98.71)** OK Analysis: This unit correctly implements the recursive formula for calculating the Eulerian number 'a(n, m)' when neither of the previous conditions ('m < 0 or m >= n' or 'm == 0') is satisfied. The logic aligns with the requirement, as it uses the correct recurrence relation:
'(n - m) * eulerian_num(n - 1, m - 1) + (m + 1) * eulerian_num(n - 1, m)'.
This step is logically correct and adheres to the specification.

After generating a self-checked and self-corrected reasoning path, we take it as a history conversation. Note that the original program is not included in case of confusion.

**Final Synthesis.** [User] You are an expert in Python coding. Your task is to write a correct program to meet the given specification based on the previous conversation. Your program can contain several functions, but make sure the main entry point is eulerian_num.
Strict Requirement: Ensure to return the complete function wrapped in a   block! No test cases!

Specification: Write a function to find the Eulerian number a(n, m), the entry point is "eulerian_num".
assert eulerian_num(3, 1) == 4

[AI] 
```
1  def eulerian_num(n, m):
2      if m < 0 or m >= n:
3          return 0
4      elif m == 0:
5          return 1
6      return (n - m) * eulerian_num(n - 1, m - 1) + (m + 1) * eulerian_num(n - 1, m)
```



By changing the condition from 'if n == 0' to 'elif m == 0', the program is successfully corrected. Through fine-grained alignment (among the requirement in natural language, step descriptions, and the generated program), RaLU can effectively correct subtle logical errors. Moreover, by taking the previous conversations as conditions, it can better prevent the propagation of errors from previous steps.

The trajectories will be made publically available as soon as the end of anonymity, where all the cases using RaLU can be shown directly.

### A.2. CFG-Driven Logic Units of Code

Figure 6 displays an example of transforming a program into a CFG and how RaLU organizes the graph to obtain a linear sequence of logic units. To enhance interpretability, units are annotated with human-readable descriptors (e.g., "LOOP BEGIN: Process each item in list X"). These labels bridge low-level code operations with high-level problem-solving intent, priming subsequent alignment stages.

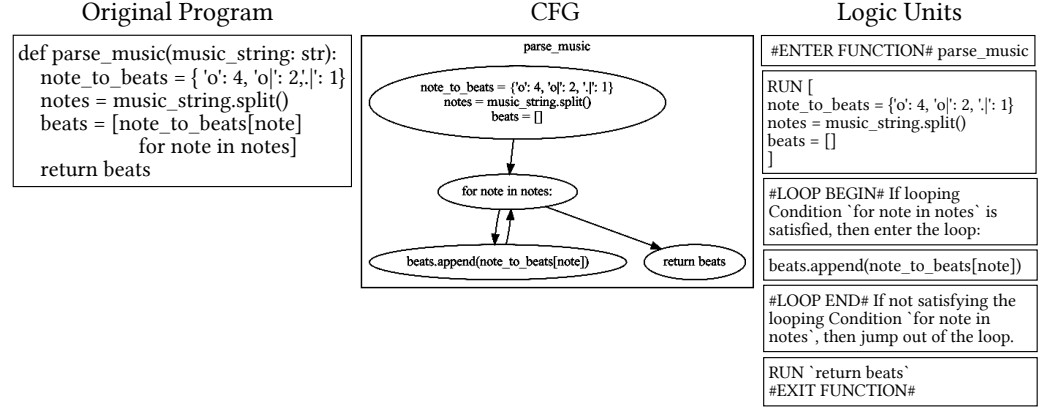

*Figure 6.* This figure depicts what a CFG looks like and how it is transformed into units. Nodes and edges in the CFG represent code blocks and control flow transitions, respectively. RaLU divides the CFG into atomic logic units at key points, with each unit labeled for better understanding.

### A.3. Condition of Applying Self-Repair

The correctness of a unit $\mathcal{U}$ involves two conditions: First, the LLM believes $\mathcal{U}$ is correct and $\mathcal{U}$ is actually correct, then we have:

$$P(\tilde{\mathcal{U}} \text{ is correct}|J = \text{OK}) = \alpha p. \tag{9}$$

Second, the probability of $\mathcal{U}$ being judged as wrong is (true negative rate plus false positive rate):

$$P(J = \text{WRONG}) = (1 - \alpha)p + \beta(1 - p). \tag{10}$$

Then, the probability of correctly repairing the unit is:

$$P(\tilde{\mathcal{U}} \text{ is correct}|J = \text{WRONG}) = \gamma_{repair} \cdot [(1 - \alpha)p + \beta(1 - p)]. \tag{11}$$

. Thus, we can rewrite $p'$ as:

$$p' = \alpha p + \gamma_{repair} \cdot [(1 - \alpha)p + \beta(1 - p)]. \tag{12}$$

To compare $p'$ and $p$, we have:

$$p' - p = -p(1 - \alpha) + \gamma_{repair} \cdot [(1 - \alpha)p + \beta(1 - p)]$$
$$= \underbrace{[(1 - \alpha)p + \beta(1 - p)]}_{P(J = \text{WRONG})} \cdot (\gamma_{repair} - \frac{(1 - \alpha)p}{(1 - \alpha)p + \beta(1 - p)}. \tag{13}$$

Note that the first term is the probability of judging the unit as WRONG so that it is always positive. The condition of $p' - p > 0$ is then transformed to:

$$\gamma_{repair} > \frac{(1-\alpha)p}{(1-\alpha)p + \beta(1-p)}. \tag{14}$$

Note that $P(\mathcal{U} \text{ is correct}|J = \text{WRONG}) = \frac{P(\mathcal{U} \text{ is correct} \wedge J = \text{WRONG})}{P(J = \text{WRONG})}$, that is:

$$P(\mathcal{U} \text{ is correct}|J = \text{WRONG}) = \frac{(1-\alpha)p}{(1-\alpha)p + \beta(1-p)}. \tag{15}$$

equal to the right part of equation 14, so the condition of $p' > p$ is $\gamma_{repair} > P(\mathcal{U} \text{ is correct}|J = \text{WRONG})$.

## B. Details of Experiment Setup

### B.1. Benchmarks

- GSM8K (Cobbe et al., 2021b) is a widely recognized benchmark to evaluate the reasoning and problem-solving capabilities of LLMs, whose name stands for "Grade School Math 8K," reflecting its focus on grade school-level math problems. The dataset contains approximately 8,500 carefully crafted math problems. Each problem in GSM8K is presented as a word problem, typically involving basic arithmetic operations (addition, subtraction, multiplication, and division) and sometimes simple algebraic concepts.

- MATH (Hendrycks et al., 2021) is proposed as a comprehensive benchmark designed to assess the mathematical reasoning capabilities of LLMs. It comprises 12,500 competition mathematics problems, which are carefully curated to cover a wide range of mathematical concepts and varying levels of difficulty. We use a subset taken from (Ling et al., 2023) named MATH-np, specifically tailored to assess the deductive reasoning skills of LLMs. It includes problems that require multi-step reasoning and the application of mathematical concepts in a structured manner.

- HumanEval (Chen et al., 2021) is a benchmark dataset to evaluate the code generation capabilities of LLMs, introduced by OpenAI. It consists of 164 hand-written Python programming problems, each with a problem specification (prompt), a predefined function signature, and a set of test cases. The primary metric used to evaluate model performance is the pass@$k$ metric, which measures the percentage of tasks for which at least one of the $k$ generated code samples passes all the test cases. Note that we only report pass@1.

- AQUA (Ling et al., 2017) is an algebraic word problem dataset collected by DeepMind, designed to evaluate the mathematical reasoning capabilities of LLMs. The questions are mainly algebraic word problems described in natural language, covering basic to medium-complexity scenarios such as profit calculation, proportion distribution, speed-time relationship, etc. There are 254 questions in the test set, and all of them are manually screened to ensure diversity and logical rigor, and their annotation quality is cross-validated.

- Mbpp (Austin et al., 2021), or Mostly Basic Python Problems, is a benchmark designed to evaluate the program synthesis capabilities of LLMs, consisting of over 900 Python programming tasks, whose problems share the same structure with that of HumenEval. It covers a wide range of basic to moderately complex Python programming problems.

- HumanEval+/Mbpp+ (Liu et al., 2023b) come from EvalPlus, a rigorous evaluation framework designed to assess the performance of LLMs in code generation by expanding the test cases of well-known benchmarks such as HumanEval and MBPP. It also maintains a leaderboard to track and compare the performance of various LLMs.

### B.2. Baselines

In our experiments, we reproduce the baselines strictly following their released code and prompts.

- Chain-of-Thought (CoT) (Wei et al., 2022) involves instructing the model to "think step by step" before arriving at a final answer. It enhances the reasoning capabilities of LLMs by explicitly guiding them to break down complex problems into a series of logical, intermediate steps. This approach mimics human reasoning by decomposing a problem into smaller, manageable sub-problems and solving them sequentially.

- Thee-of-Thought (ToT) (Yao et al., 2023) enhances LLM reasoning capabilities by simulating human problem-solving strategies. ToT breaks down the problem-solving process into smaller, manageable, intermediate steps called "thoughts." For each state in the thought tree, the LLM generates multiple potential next thoughts. Each generated thought is evaluated for its potential to lead to a solution. Then, it employs search algorithms such as Breadth-First Search (BFS) or Depth-First Search (DFS) to explore the thought tree systematically. The structured nature of the thought tree makes the reasoning process more transparent and interpretable.

- Program-of-Thought (PoT) (Chen et al., 2023b) or its similar approach Program-Aided Language Model (PAL) (Gao et al., 2023) both represent a novel approach that combines the strengths of LLMs with the precision of programming languages. The LLM reads a natural language problem and generates a program as the intermediate reasoning step. They aim to decompose reasoning and computing by offloading the solution step to a symbolic interpreter, which leverages the LLM's reasoning abilities while mitigating its weaknesses in logical and arithmetic operations. PoT or PAL will degrade to direct prompting in the face of program generation tasks.

- Self-Consistency (SC) (Wang et al., 2023) is a decoding strategy designed to improve the accuracy and reliability of reasoning processes. It involves generating multiple reasoning paths for a given problem and selecting the most consistent answer among them (majority voting). The consistency can be directly computed (for numerical calculation tasks) or determined by LLMs (either the same LLM or another LLM).

- Self-Calibration (SCal) (Kadavath et al., 2022) an advanced prompting technique designed to enhance the accuracy and reliability of LLMs by enabling them to evaluate their own outputs. The LLM generates an initial answer to a given question, and it is prompted to assess the correctness of its own response.

- Self-Refine (SR) (Ranaldi & Freitas, 2024) is an iterative refinement technique designed to enhance the output quality of LLMs by incorporating self-generated feedback. Specifically, the LLM generates an initial response to a given prompt, and the same LLM evaluates the initial output and provides actionable feedback, identifying areas for improvement. With the feedback, the same LLM refines the initial output, aiming to improve its quality. This response-feedback-refine pipeline can be repeated multiple times until the output meets a predefined stopping criterion.

- Self-Debugging (SD) (Chen et al., 2024b) is an innovative technique designed to enable LLMs to identify and correct errors in the code they generate without requiring additional model training or human intervention. This method is inspired by the "rubber duck debugging" technique used by human programmers, where explaining code line-by-line in natural language helps identify and fix errors. Since it targets program bugs, it cannot be directly applied to mathematical reasoning tasks.

- Self-Check (SCheck) (Miao et al., 2024) is a prompting technique that enables LLMs to evaluate their own reasoning and identify errors in their step-by-step solutions. It first provides several step-by-step solutions through CoT prompting. Then, it identifies the relevant context and target for each step in its reasoning process. Afterward, the LLM generates an independent alternative step based on the extracted context. The original step is compared with the regenerated alternative. If they match, the original step is deemed correct. The reasoning path with the most "correct" steps will be selected (weighted majority voting).

## C. Additional Comparisons

### C.1. RaLU v.s. Closed-Source LLMs

As shown in Table 5, in cross-domain benchmarks (mathematical reasoning and code generation), RaLU exhibits better reasoning capabilities to mainstream closed-source models (i.e., GPT-4o, GPT-4-Turbo, and Claude-Sonnet-3.5) and significantly outperforms GPT-3.5-Turbo (+38.16% on average).

RaLU achieves the highest scores on the extended versions of code generation benchmarks (HumanEval+/Mbpp+), despite its slightly lower performance on original HumanEval/MBPP ($\Delta$=-2.24%). This inversion reveals a critical insight that unit-level correction benefits to solving multi-constraint tasks, since the augmented test suites introduced by the plus version of benchmarks require models to simultaneously satisfy competing constraints. While closed-source models often overfit to dominant patterns in pretraining data, RaLU's unit-level rewind mechanism enables iterative constraint alignment. This explains the 9.46% improvement on mathematical reasoning benchmarks where LLMs struggle with numerical computation.

*Table 5.* RaLU using open-sourced LLMs (DeepSeek-V3 or Qwen2.5-72B) can even outperform state-of-the-art closed-source LLMs, whose results are reported by previous studies. It achieves SoTA results on both mathematical reasoning benchmarks by making an improvement of 9.46%, though there are no results on o1 models so we have to omit them. RaLU also enhances code generation by 2.61% on the plus versions of benchmarks compared to o1-preview, despite the slight decline (-2.24%) on the standard benchmarks.

| Dataset | o1 Preview | o1 Mini | GPT-4o | GPT-4-Turbo | Claude-3.5-Sonnet | GPT-3.5-Turbo | Ours (RaLU) |
|---|---|---|---|---|---|---|---|
| GSM8K | 0.969 | 0.951 | 0.948 | 0.926 | 0.950 | 0.822 | **0.980** |
| MATH-np | - | - | 0.697 | 0.590 | 0.623 | 0.347 | **0.821** |
| HumanEval | **0.963** | **0.963** | 0.927 | 0.902 | 0.872 | 0.835 | 0.939 |
| HumanEval+ | 0.890 | 0.890 | 0.872 | 0.866 | 0.817 | 0.707 | **0.902** |
| Mbpp | **0.955** | 0.931 | 0.876 | 0.857 | 0.894 | 0.825 | 0.937 |
| Mbpp+ | 0.802 | 0.788 | 0.722 | 0.733 | 0.743 | 0.697 | **0.833** |

## C.2. RaLU on a smaller model

To demonstrate the effectiveness of RaLU on a smaller model, we supplemented experiments with Qwen2.5-14B on MATH and Mbpp/Mbpp+. As shown in Table 6, RaLU still provides significant improvements for smaller yet capable models.

*Table 6.* RaLU achieves the best performance with Qwen-14B-Instruct on Mbpp/Mbpp+ and MATH-np, showing its generalizations among different scales of LLMs.

| Dataset | Direct | CoT | ToT | PoT/SR* | SC | SCal | SCheck/SD | RaLU | ΔGain(%) |
|---|---|---|---|---|---|---|---|---|---|
| Mbpp | 0.840 | 0.860 | 0.831 | 0.804 | 0.868 | 0.852 | 0.852 | **0.902** | +3.917 |
| Mbpp+ | 0.725 | 0.733 | 0.720 | 0.698 | 0.754 | 0.706 | 0.714 | **0.839** | +11.273 |
| MATH-np | 0.603 | 0.691 | 0.651 | 0.731 | 0.751 | 0.710 | 0.593 | **0.784** | +4.394 |

# D. Case Studies

## D.1. Reasoning hallucinations in CoT-PoT-integrated method (single reasoning path)

**[User]** You are an expert in coding. Your goal is to write a program to meet the given specification. Let's think it step by step. Show your thinking process explicitly. Example Response:

def add_nums(a, b):
return a+b

Analysis: Using operation '+' to directly implement the requirement of returning the sum of two numbers.

**Strict Requirement**: Ensure to return the complete function wrapped in a   block! No test cases!

Specification: In this task, you will be given a string that represents a number of apples and oranges that are distributed in a basket of fruit this basket contains apples, oranges, and mango fruits. Given the string that represents the total number of the oranges and apples and an integer that represent the total number of fruits in the basket. Return the number of the mango fruits in the basket.
For example:
fruit_distribution("5 apples and 6 oranges", 19) →19 - 5 - 6 = 8
fruit_distribution("0 apples and 1 oranges",3) →3 - 0 - 1 = 2
fruit_distribution("2 apples and 3 oranges", 100) →100 - 2 - 3 = 95
fruit_distribution("100 apples and 1 oranges",120) →120 - 100 - 1 = 19

**[AI (Deepseek V3)]** Let's break down the problem step by step:
1. **Input Analysis**:

- The input is a string 's' that contains the number of apples and oranges in the format 'X apples and Y oranges'.
- The second input is an integer 'n' representing the total number of fruits in the basket.

2. **Objective**:
- We need to extract the number of apples ('X') and oranges ('Y') from the string 's'.
- Then, calculate the number of mangoes by subtracting the sum of apples and oranges from the total number of fruits 'n'.

3. **Approach**:
- Use string manipulation to extract 'X' (apples) and 'Y' (oranges) from the string.
- Convert these extracted values to integers.
- Subtract the sum of apples and oranges from 'n' to get the number of mangoes.

4. **Implementation**:
- Split the string into parts using spaces or specific keywords like "apples" and "oranges".
- Extract numerical values corresponding to apples and oranges.
- Perform arithmetic operations to find mangoes.

Here's how we can implement this:

```
1   def fruit_distribution(s, n):
# Split the string into parts
2       parts = s.split()
# Extract apple count
3       apple_count = int(parts[0])
# Extract orange count
4       orange_count = int(parts[2])
# Calculate mango count
5       mango_count = n - apple_count - orange_count
6       return mango_count
```


Analysis:
- The function splits the input string into parts using spaces.
- It extracts apple count as an integer from index 0.
- It extracts orange count as an integer from index 2.
- Finally, it calculates mango count by subtracting apple and orange counts from total fruits 'n'.

The NL reasoning correctly emphasizes extracting numerical values for apples (X) and oranges (Y) from the input string. However, the code fails to parse the string correctly. For example, for an input "5 apples and 6 orange" in a standard test case, splitting by spaces produces a list of [5, apples, and, 6, oranges]. The code erroneously takes parts[2] (value "and") for oranges, which is not a numerical value. This causes a ValueError when converting "and" to an integer.

Instead, RaLU correctly fixes this error during the self-judging and self-correcting by responding with:

**[AI (Deepseek V3)]** WRONG
<Fix>
RUN [
parts = s.split()
apples = int(parts[0])
oranges = int(parts[3])
mangoes = n - apples - oranges
return mangoes
]
</Fix>

Analysis: The step incorrectly assumes that the oranges count is always at index 2 of the split string. However, in the input string format, the oranges count is at index 3. The correction adjusts the index to 3 to correctly extract the number of oranges. This aligns with the specification, which requires extracting the number of apples and oranges from the string and calculating the number of mangoes.

## D.2. Reasoning hallucinations in CoT-step-driven RaLU (ablation study)

**[User]** You are an expert in solving math questions. Your goal is to return the final answer to solve the given question and show your thinking process explicitly. Let's think it step by step. For example:
Question: John has 10 apples. He gives away 4 and then receives 5 more. How many apples does he have?
Response:
<Step>1: John starts with 10 apples.</Step>
<Step>2: He gives away 4, so 10 - 4 = 6.</Step>
<Step>3: He then receives 5 more apples, so 6 + 5 = 11.</Step>
<Answer>11</Answer>

Question: Find the integer that lies between $\sqrt[3]{-45}$ and $\sqrt[3]{-101}$."

**[AI (Llama3.3)]**
<Step>1: The expression $\sqrt[3]{-45}$ is equal to $-3$, because $(-3)^3 = -27$. Therefore, $-4 < \sqrt[3]{-45} < -3$.</Step>
<Step>2: The expression $\sqrt[3]{-101}$ is equal to $-5$, because $(-5)^3 = -125$. Therefore, $-6 < \sqrt[3]{-101} < -5$.</Step>
<Step>3: Comparing the two intervals, $-5 < -4$, we can see that the integer that lies between $\sqrt[3]{-45}$ and $\sqrt[3]{-101}$ is $-4$ and $-5$.</Step>
<Answer>-4 or -5</Answer>

After several turns of self-checking and self-correcting the steps, this LLM writes a program to solve this problem:

**[AI (Llama3.3)]** 
```
# We know that -45 and -101 are both negative, so their cube roots will also be negative.
# To find the integer between these two cube roots, we first need to find the cube roots themselves.
# Calculate the cube root of -45 and -101
1    root45 = round(-45 ** (1/3.0))
2    root101 = round(-101 ** (1/3.0))
# Now, we need to find the integer between these two roots.
# Since both roots are negative, we can use the min function to get the one that's closer to zero
3    ans = max(root45, root101)
4    print(ans)
```


Given the execution result of its written program, the LLM responds as follows:

**[AI (Llama3.3)]** Based on the given program and the calculated 'ans', I will show my thinking process explicitly. Since we are trying to find the integer between $\sqrt[3]{-45}$ and $\sqrt[3]{-101}$, we first calculated the cube roots themselves using the program. The results are 'root45 = -3' and 'root101 = -5'. Now, we need to find the integer between these two roots. Since both roots are negative, we can use the 'max' function to get the one that's closer to zero. The program calculated 'ans = max(root45, root101) = max(-3, -5) = -3'. However, we need to find the integer that lies between $\sqrt[3]{-45}$ and $\sqrt[3]{-101}$. Since '-3' is the cube root of '-45', we need to add 1 to it to get the integer that lies between the two cube roots. Therefore, the final answer is:<Answer>-3 + 1 = -2</Answer>

Though the generated code correctly calculates $\sqrt[3]{-45} \approx -3.56$ and rounds it to $-4$, and $\sqrt[3]{-101} \approx -4.66$ rounded to -5, it still misinterprets the code's output as $-3$ leading to an invalid adjustment $-3 + 1 = -2$. This is because in the original CoT steps, the LLM incorrectly state that $\sqrt[3]{-45} \approx -3$, which is mathematically flawed. In the final response, it conflates the code's output $(-4)$ with the incorrect initial NL reasoning $(-3)$, creating a disjointed argument. This misconnection invalidates the final answer. Additionally, the final NL explanation introduces an extraneous step: "we need to add 1 to it." This is not mathematically justified, as the correct answer has already been computed by "ans = max(root45, root101)". The addition of 1 is a hallucinated step.

