# OpenReview forum: "Reasoning-as-Logic-Units: Scaling Test-Time Reasoning in Large Language Models Through Logic Unit Alignment"
_ICML.cc/2025/Conference — ICML 2025 poster_

### Official Review · Reviewer_dezw · 2025-02-26

**Overall Recommendation:** 3

**Summary:**

This paper introduces Reasoning-as-Logic-Units (RaLU), a novel test-time reasoning framework designed to address hallucinations in LLM reasoning and enhance their performance in mathematical and coding reasoning tasks.

Specially, RaLU consistes of three parts.
Logic Unit Extraction: begins by generating an initial program and constructing a Control Flow Graph (CFG) using static code analysis. The CFG is then decomposed into discrete logic units, each representing a self-contained computational intent. This decomposition allows for structured refinement of the program's logic.

Logic Unit Alignment: engages in iterative dialogue with the LLM to judge, explain, and correct each logic unit. If a unit is incorrect, it is refined and re-evaluated. This process continues until all units are validated or a predefined threshold is reached.

Solution Synthesis: synthesizes them into a coherent reasoning path and generates the final solution. This ensures that the final solution inherits the logical rigor of the program while retaining the interpretability of natural language.

The authors conduct experiments on several mathematical reasoning and code generation datasets.

**Claims And Evidence:**

RaLU significantly reduces reasoning hallucinations
- RaLU outperforms existing baseline methods
- The authors give error analysis where the proposed method identifies and corrects the errors.
- It also give a theoratical explanation.

RaLU enhances the accuracy and interpretability
- It achieves good performance on each benchmark.
- It can generate readable explanations, improving transparent and interpretability.

**Essential References Not Discussed:**

No

**Experimental Designs Or Analyses:**

The experimental designs is reasonable.

**Methods And Evaluation Criteria:**

Yes

**Other Comments Or Suggestions:**

See above

**Other Strengths And Weaknesses:**

See above

**Questions For Authors:**

No

**Relation To Broader Scientific Literature:**

The main focus of this paper is to organize and plan the sub-steps of multi-step reasoning through prompt engineering. From this perspective, the proposed method is relatively similar to previous approaches.

The difference lies in the fact that this paper uses logical units as interpretable and verification units for reasoning, thereby introducing new elements to the control flow of reasoning.

**Theoretical Claims:**

Yes, I have checked

---

> ### Author Rebuttal · Authors · 2025-04-01
>
> Thank you for the review. To ensure we address your concerns with precision, could you kindly clarify your suggestions? We are delighted to address any questions you may have and refine our paper accordingly. We look forward to your feedback and reassessment.

---

### Official Review · Reviewer_jMQo · 2025-03-11

**Overall Recommendation:** 4

**Summary:**

This paper presents a novel test-time scaling framework, Reasoning-as-Logic-Units (RaLU), which consists of three main steps: Logic Unit Extraction (directly generating a program to address the given problem, and using static analysis tools to create a control flow graph to decompose the program into logic units),  Logic Unit Alignment (iteratively judge and refine each unit), and Solution Synthesis (generate the final answer according to the conversation history). The authors compare their RaLU with other previous methods on several benchmarks in mathematical reasoning (GSM8K, MATH) and algorithmic reasoning (HumanEval+, MBPP+) across different open-sourced base models (DeepSeekV3,  Llama3.3-70B-Instruct and Qwen2.5-72B-Instruct), and RaLU achieves consistently the best accuracy. They also conducted ablation studies, comparing with line-by-line and NL-step, to demonstrate the effectiveness of decomposing the program into logic units and using program-aided techniques.

**Claims And Evidence:**

Yes

**Essential References Not Discussed:**

No

**Experimental Designs Or Analyses:**

I checked the details in the main text, which look sound to me.

**Methods And Evaluation Criteria:**

Yes

**Other Comments Or Suggestions:**

1. The format of equation (4) is wrong.
2. In (6), should $1-\beta$ be replaced with $\beta$?

**Other Strengths And Weaknesses:**

Strengths:

1. RaLU achieves great performance in several benchmarks such as GSM8k, MATH, HumanEval+, MBPP+.

2. The ideas of decomposing complex reasoning into unit steps,  using both program and natural languages to reduce reasoning hallucinations, and enforcing more rigorous and interpretable reasoning steps, and using external tools (creating CFG) are all great and effective.

3. The paper is well written and structured and the ideas and methods are clearly presented.

Weaknesses:

1. Since RaLU is a test-time scaling method, it would be better to also compare the inference cost (e.g., total time cost, token consumptions) with previous methods to see the computation-accuracy tradeoff.

**Questions For Authors:**

Why in (4) and (5), the probabilities of each token are added instead of multiplied (or equivalently, the log of probabilities are added)?

**Relation To Broader Scientific Literature:**

The paper presents a pioneering test-time scaling framework designed to tackle reasoning hallucinations and enhance the reasoning capabilities of LLMs.

**Theoretical Claims:**

N/A

---

> ### Author Rebuttal · Authors · 2025-04-01
>
> We are sincerely grateful for your recognition of RaLU’s contributions. Many thanks to your constructive comments to enhance our work.
>
> # Questions
> We appreciate the reviewer's insightful question.
> Let’s use perplexity=exp(-1* Mean(token probabilities)) as another metric using “multiplying”. We have supplemented an ablation study, which revealed that the impact of the selection strategy is marginal. Specifically, we conducted experiments using Qwen-72B-Instruct on the MATH dataset because the MATH dataset is complex enough, and there are enough cases (89/700) where the branch reaches the threshold, and different selection strategies may have a relatively significant impact on the final result.
> We used three comparison strategies: random selection, choosing the candidate with the minimum perplexity, and choosing the last one. The accuracy results with the 89 cases are as follows.
>
> |Confidence(original)|Random|Perplexity|Last|
> |-|-|-|-|
> |42/89=0.472|40/89=0.449|45/89=**0.506**|38/89=0.427|
>
> This can be attributed to two key factors:
> - During self-correction iterations, the LLM tends to produce tokens with consistently high probabilities (the average probability >0.9 for most responses in our analysis). This results in minimal variance between the average probability (our confidence score) and perplexity (geometric mean equivalent) metrics - their Pearson correlation reaches 0.93. Essentially, both metrics reflect similar confidence patterns.
> - Through qualitative analysis, we observed that many candidate units generated within budget limits are functionally equivalent variants differing only in implementation details. This semantic equivalence explains why random selection only causes minor performance degradation.
>
> While perplexity-based selection slightly outperforms our original strategy (+0.9%), the limited gain suggests that the verification-revision loop in Stage 2 already filters out most critical errors before selection occurs. It also indicates that multiplying can be a insightful metric and deserves consideration. In the revised appendix, we will add relevant discussions and provide different candidate selection strategies in the code to accommodate various scenarios. Thank you again for helping us enhance this work.
>
> # Strengths And Weaknesses
> **Inference cost**: We appreciate the reviewer's insightful comment. We list the average token consumption of RaLU and baselines on MATH-np using Qwen-72B-Instruct. RaLU consumes 15x tokens compared to CoT, while saving 10x tokens compared to multi-path reasoning baselines such as Self-Check and ToT. We will include the information about token cost in Appendix.
> | Token | Direct   | CoT   | ToT     | PoT | SC    | SCal  | SCheck     | RaLU  |
> | -  | - | - | - | - | - | - | - | - |
> | Input | $8 \times 10^{4}$ | $8 \times 10^{4}$ | $7.5 \times 10^{7}$ | $2 \times 10^{5}$ | $1.3 \times 10^{6}$ | $1.5 \times 10^{5}$ | $2 \times 10^{7}$ | $4.8 \times 10^{6}$ |
> |Output| $7 \times 10^{4}$ | $1.5 \times 10^{5}$ | $1.2 \times 10^{7}$ | $9 \times 10^{4}$ | $3 \times 10^{6}$ | $8 \times 10^{4}$ | $1.4 \times 10^{6}$ | $7 \times 10^{5}$
> | -  | - | - | - | - | - | - | - | - |
>
> # Other Comments
> 1. Thanks for pointing out the error of equation (4). We correct it as $\text{Conf}(\tilde{\mathcal{U}}) = \frac{1}{n}\sum_{j=1}^{n} \sigma(lp_j) $
>
> 2. Thank you very much for pointing out our clerical error. In the Appendix, $1-\beta$ also needs to be replaced with $\beta$, and the rest of the derivation process and conclusions remain unchanged. Thank you very much for your meticulous review.

---

### Official Review · Reviewer_KWVa · 2025-03-14

**Overall Recommendation:** 3

**Summary:**

This paper introduces a novel prompt engineering-based approach, Reasoning-as-Logic-Units (RaLU), which consists of three key components:1) Logic Unit Extraction, 2) Logic Unit Alignment, and 3) Solution Synthesis, to enhance the reasoning capability of the LLMs.
RaLU decomposes the task into multiple logic units, by aligning the code statements within each logic unit with the natural language task specification, it mitigates reasoning hallucinations in the model, and then it synthesizes all the logic units to generate the final solution.
Experimental results have demonstrated that RaLU outperforms existing baselines on benchmarks for both mathematical and code generation tasks

**Claims And Evidence:**

1. Most of the claims are well evaluated. However, there is a little concern in Section 4.2, RaLU v.s. Self-Correction Methods. “Many existing self-correction-based methods (e.g., Self-Refine and SelfDebug), often degrade performance by introducing errors into initially correct responses–a flaw exacerbated by their assumption of imperfection existence in the initial response attempt.” I am not sure whether this assumption is correct, more evidence should be given. Additionally, does the baseline (SCheck/SD⋆) used in this paper also adopt this assumption?

2. The initially generated program largely determines the quality of the final result, as all logic units are extracted from it. Therefore, experiments should be conducted multiple times to reduce the impact of randomness.

3. In the process of judging the correctness of each logic unit, if the limit budget is reached, the method chooses the candidate with highest confidence score as the correct one.  There should be an experiment by comparing it with another strategy, e.g., choosing a candidate randomly.

**Essential References Not Discussed:**

The paper ‘ChatRepair’ has been referred to in this paper, but the discussion about it is not enough. The method proposed in this paper relies entirely on the model's intrinsic reasoning ability, whereas ChatRepair serves as a representative approach that leverages external feedback to guide reasoning. On the benchmark of code generation, it would be beneficial to include ChatRepair as a baseline for comparative evaluation. Otherwise, a more detailed discussion should be provided.

**Experimental Designs Or Analyses:**

Overall, the experimental design is reasonable. However, some issues remain, such as the design of the ablation study, the validation of the effectiveness of certain procedures, and the impact of randomness.

**Methods And Evaluation Criteria:**

Yes. The methods make sense for enhancing the reasoning ability for LLMs to perform better on both mathematical and code generation tasks.

**Other Comments Or Suggestions:**

N.A.

**Other Strengths And Weaknesses:**

The approach is novel by integrating the CFG and extracting the logic units from it to align the code statements within each logic unit with the natural language task specification. And experimental results have demonstrated that RaLU outperforms existing baselines on the chosen benchmarks.

Potential Weakness:
1. Experiments should be conducted with multiple random seeds.
2. No specific experiments can demonstrate the effectiveness of selecting the candidate with the highest confidence score.

**Questions For Authors:**

1. Can you show the performance with running multiple times to remove the potential risk of randomness?
2. Can you conduct specific experiments to demonstrate the effectiveness of selecting a candidate with the highest confidence score? e.g., comparing with the random selection.
3. How about the performance of ChatRepair on the benchmarks of code generation?

**Relation To Broader Scientific Literature:**

This paper presents an effective approach to enhance model reasoning capabilities by decomposing a task into multiple parts or breaking it down into several sub-tasks and solving them sequentially. This approach can be considered a general framework that can be applied to a wide range of code-related tasks.

**Theoretical Claims:**

The theoretical claims in this paper are reasonable.

---

> ### Author Rebuttal · Authors · 2025-04-01
>
> Thank Reviewer KWVa for your constructive feedback on our work. We have carefully considered all comments and hope our point-by-point response can address your questions.
>
> ## Questions
>
> 1. **Multiple runs of experiments**: Thank you for your feedback! Given the inherent stochastic nature of LLMs and the absence of seed support in the official API, we additionally conducted two independent trials on Qwen-72-Instruct for MATH and Mbpp/Mbpp+ benchmarks (temperature=0.7, identical prompting strategies) to further validate the robustness of RaLU under different model configurations. While these preliminary results (reported as mean ± std in Table 2) align with our prior findings, we will finalize three full replications across all models and datasets before publication to ensure statistical rigor. This iterative process strengthens our confidence in RaLU's consistent performance across varying model architectures and task domains.
> | Benchmark | Direct       | CoT          | ToT          | PoT/SR\*     | SC           | SCal         | SCheck/SD    | RaLU             |
> | - | - | - | - | - | - | - | - | - |
> | Mbpp      | 0.923±0.0070 | 0.895±0.0014 | 0.905±0.0045 | 0.860±0.0046 | 0.922±0.0033 | 0.924±0.0038 | 0.905±0.0038 | **0.957±0.0012** |
> | Mbpp+     | 0.788±0.0021 | 0.761±0.0052 | 0.772±0.0046 | 0.725±0.0038 | 0.779±0.0061 | 0.787±0.0122 | 0.750±0.0349 | **0.856±0.0017** |
> | Math-np   | 0.705±0.0065 | 0.701±0.0085 | 0.695±0.0045 | 0.743±0.0223  | 0.764±0.014  | 0.725±0.0065 | 0.685±0.0088 | **0.803±0.0085** |
>
> 2. **Ablation Study of Candidate Unit Selection**: Thank you for your insightful suggestion. We conducted ablation experiments using Qwen-72B-Instruct on MATH,  which is sufficiently complex with enough cases (89/700) where the branch reaches the threshold. We used three comparison strategies: random selection, choosing the candidate with the minimum perplexity, and choosing the last one. The accuracy results are as follows.
>
> | Confidence (original) | Random      | Perplexity      | Last        |
> | - | - | - | - |
> | 42/89=0.472           | 40/89=0.449 | 45/89=**0.506** | 38/89=0.427 |
>
> The results revealed that the impact of the selection strategy is marginal. During self-correction iterations, LLMs tend to produce tokens with high probabilities. This results in minimal variance between the average probability (confidence) and perplexity (geometric mean equivalent) metrics. Through qualitative analysis, we observed that many candidate units generated within budget limits are functionally equivalent variants differing only in implementation details. This semantic equivalence explains why random selection only causes minor performance degradation.
>
> Due to the limited space, we provide a more detailed description in our response to Reviewer jMQo, Question Part, for your reference.
>
> 3. **ChatRepair Comparison**: Thank you for your suggestion. We will discuss the differences between ours and ChatRepair-like approaches more comprehensively. As you mentioned, the core design concepts of the two are distinct. Our approach relies entirely on the model's intrinsic reasoning capabilities, whereas ChatRepair depends on external feedback for corrections.
> In practice, it is challenging to obtain comprehensive test cases. ChatRepair also struggles with complex mathematical reasoning and may cause overfitting and data leakage using test data. As a feedback-based correction method, ChatRepair does not sufficiently align with our method in terms of problem assumptions, input constraints, and evaluation metrics. Our method has been comprehensively compared with relevant baselines for test-time reasoning enhancement. The baseline papers also did not compare against ChatRepair. Also,  ChatRepair is an Automated Program Repair tool that aims to generate patches for buggy programs instead of code generation or reasoning.
>
> ## Claims
> 1. **RaLU v.s. Self-Correction Methods**: We appreciate the reviewer's insightful feedback regarding our analysis of self-correction methods. We will make the expressions more rigorous by changing it to “Many existing self-correction-based methods (e.g., Self-Refine, Self-Debug) implicitly encourage differences between self-corrected responses and initial responses, which can potentially introduce errors into initially correct responses.”
> According to their original papers, Self-Refine and Self-Debug prompt the LLM to fix the response based on self-generated or external feedback and adopt the newly generated response as the final result. On the other hand, Self-Check generates multiple candidate steps after information extraction and other procedures. It then decides on step-through voting, maintaining the possibility of retaining the original response. In Self-Check, different candidate responses are treated equally. Our method also allows retaining the original response to mitigate the issue of introducing errors into a correct response.

---

> > ### Comment · Reviewer_KWVa · 2025-04-08
> >
> > I appreciate the authors' efforts in addressing my comments, including the new results and clarifications. I would like to increase my score.

---

> > > ### Author Response · Authors · 2025-04-08
> > >
> > > Thank you for your kind words and willingness to increase the score! We greatly value your constructive feedback, which has been instrumental in improving our work.

---

### Official Review · Reviewer_4xYr · 2025-03-15

**Overall Recommendation:** 4

**Summary:**

The paper proposes a novel prompting/structured reasoning technique method (RaLU) that mitigates reasoning inconsistencies within the generated LLM output by proposing an alignment (alignment between the task and the generated code) and self-refinement (decomposing code into logical units and iteratively refining in context with LLM judges) modules for structures and decomposable reasoning. The results of the paper show a SOTA performance with increases on 4 math and algorithmic reasoning tasks. Further empirical proofs show that structuring the reasoning process with initial code and decomposing the code with CFG is essential for achieving strong performance.

**Claims And Evidence:**

All of the empirical claims are well supported by evaluating the method vs diverse benchmarks that include CoT reasoning or variations (ToT), structured reasoning (PoT) and self-refinement. The method shows performance gains across 3 different models compared to all of the tested techniques. Further ablations are also consistent, showing that key components of the method (Decomposition with CFG and writing code) are necessary for the reasoning paradigm.

With this said, the theoretical explanation and claims seem either to be overstating the contribution components of the method or are handwavy (Section 3 mainly):

1) While the identified three types of reasoning inconsistencies seem correct, how can we be sure that those modes are exhaustive or cover the majority of reasoning inconsistency types? Is there any analysis w.r.t. the identified/presented error types?

2) Across the method explanation (both in the intro and section 3), the authors repeatedly mention that the solution synthesis (final step) results in "verified" (each node is verified) reasoning paths. However, as all of the units are judged and refined with an LLM and are not deterministically confirmed to be true, the statement seems to be a tad strong. (Further questions and concerns can be explored in the Theoretical claims section)

3) While the LLM-judge approach has shown some performance yields, it has also been shown [1,2,3] that there is a self-preference bias within LLMs, which is amplified during the process of self-refinement. How do the self-refinement and self-judging modules of RaLU stack against this phenomenon? Has it been tested?

4) After Judging and refining all of the segmented units, they are recombined (final synthesis) into a new program. However, after individual refinement, the units are aligned with the task, yet are they aligned and consistent with respect to each other? Can we just concatenate the refined chunks of code and obtain an executable program? Does the code after the final synthesis step explicitly and verifiably include all of the chunks from the refined units?

5) How does the written CFG heuristic segment deeply nested loops and branches? How can varying levels of nesting be recombined after refinement? Are those decisions left to the LLM?


[1] Panickssery, A., Bowman, S.R., Feng, S. (2024). LLM Evaluators Recognize and Favor Their Own Generations. arXiv preprint arXiv:2404.13076.

[2] Xu, W., Zhu, G., Zhao, X., Pan, L., Li, L., Wang, W.Y. (2024). Pride and Prejudice: LLM Amplifies Self-Bias in Self-Refinement. arXiv preprint arXiv:2402.11436.

[3] Wataoka, K., Takahashi, T., Ri, R. (2024). Self-Preference Bias in LLM-as-a-Judge. arXiv preprint arXiv:2410.21819.

**Essential References Not Discussed:**

N/A

**Experimental Designs Or Analyses:**

The ablations and experimental designs are sound.

**Methods And Evaluation Criteria:**

The chosen datasets, benchmarked methods and evaluations are relevant to the task explored in the paper. Although the choice of the models and the datasets is not explicitly systematic or weel justified. (2 additional questions w.r.t. this are in the questions section)

**Other Comments Or Suggestions:**

The Bayesian explanation/argument (both in the main paper and appendix) seems rather handwavy and makes very strong assumptions w.r.t. the self-judging and self-refinement processes (Questions and concerns above). Please either add more mathematical rigour and mention the assumptions explicitly or replace the section.

**Other Strengths And Weaknesses:**

N/A

**Questions For Authors:**

1) Why are these sets of varying models chosen exactly? How does the idea scale for models <=30B and 100B+ (outside of deepseek)
2) Can the authors mention what is the average token (amount) difference between CoT, PoT and RaLU? How efficient is it to use RaLU?
3) Can the authors provide additional benchmarks that would show the consistency of the method? GSM8k, MATH and HumanEval well-saturated benchmarks. Benchmarks such as AQUA, ProofWriter, AR-LSAT etc. might be better suited for testing the method.

**Relation To Broader Scientific Literature:**

The paper contributes a novel prompting/structured reasoning method for tackling complex reasoning tasks with controllable modules for decomposition, task alignment and self referential improvement.

**Theoretical Claims:**

The main theoretical claim of the paper is a "Bayesian" argument about the generated self-refined (self-repair) unit (and subsequently a solution) is more likely to be correct than the unrefined counterpart. Both of these arguments (Section 3.2-3.3) rest heavily on the assumptions of optimality of the LLM judge and the LLM refinement process. The Bayesian argument only works in case the expected performance of the Judge is not random and is even much better than the judgement/generation of the model that outputted the initial program. Given that the Judge and the initial generator are the same LLM with different contexts, I cannot really consider the Bayesian lens argument to have full mathematical rigour.

---

> ### Author Rebuttal · Authors · 2025-04-01
>
> We are grateful for your valuable comments and hope this response can address your questions.
>
> # Questions
> 1. We select the latest models from three renowned open-source families. The effectiveness of RaLU is not directly tied to the model's size but rather depends on the model's reasoning capabilities and the ability to follow detailed instructions. We supplemented experiments with Qwen-14B on MATH and Mbpp/Mbpp+. As shown in the Table, RaLU still provides significant improvements for smaller yet capable models.
> |Benchmark|Direct|CoT|ToT|PoT/SR\*|SC|SCal|SCheck/SD|RaLU|
> |-|-|-|-|-|-|-|-|-|
> |Mbpp|0.840|0.860|0.831|0.804|0.868|0.852|0.852|**0.902**|
> |Mbpp+|0.725|0.733|0.720|0.698|0.754|0.706|0.714|**0.839**|
> |MATH-np|0.603|0.691|0.651|0.731|0.751|0.710|0.593|**0.784**|
>
> 2. Due to the limited space, please refer to our response to **Reviewer jMQo, Strengths And Weaknesses**
>
> 3. We have added AQUA on Qwen-72B-Instruct. Impressively, our RaLU framework continues to achieve the best performance. We agree that more benchmarks would further validate RaLU's robustness and plan to include AQUA in our revised version.
>
> |Benchmark|Direct|CoT|ToT|PoT|SC|SCal|SCheck|RaLU|
> |-|-|-|-|-|-|-|-|-|
> |AQUA|0.764|0.799|0.791|0.807|0.779|0.811|0.772|**0.846**|
>
> # Claims
> 1. Thank you for your insights on "reasoning hallucinations." Here, they primarily refer to the logic mismatch between code- and NL-based reasoning—the key challenge RaLU addresses. We do not cover traditional logical inconsistencies (e.g., reasoning-final answer mismatch) and will clarify this in the revised paper.
>
> Given the difficulty in exhaustively categorizing reasoning errors, we adopt a top-down approach: abstracting reasoning hallucinations as disruptions to one-to-one sequence mappings (e.g., "12345"$\leftrightarrow$ "abcde"). These fall into three core types:
> - Element errors
> - Missing/redundant elements
> - Sequence misordering
> Other errors can be viewed as combinations of these. (Note: We’ll add "or vice versa" to "1) accurate NL step…").
>
> We acknowledge the theoretical challenge of covering all error types due to real-world complexity. If they are hallucinations independent of the three types, we'll supplement experiments to further analyze whether RaLU can mitigate the new type. Our classification aims to guide technical improvements—even if marginal types exist, solving these three significantly boosts reliability, with extensible methodology.
>
> 2. We will replace "verified" with "self-verified" to enhance the rigor of the text, emphasizing that this is an internal validation process only relying on the LLM itself to prevent such ambiguity.
>
> 3. Although the LLM in RaLU may misjudge a unit, this is not a self-preference bias. Self-preference bias occurs when the LLM favors answers aligned with its training distribution, deviating from human preferences [1]. In RaLU, self-judgment only assesses correctness—misclassifying an incorrect unit as correct is an error (not a bias), affecting its final accuracy, not human-aligned processes.
> A more relevant issue might be confidently incorrect predictions, where the LLM overestimates its answers' correctness. To mitigate this, RaLU separates program generation and judgment/refinement into different dialogues, obscuring the units' source. Additionally, units are extracted from generated programs with modified indicators, reducing overconfidence in familiar distributions.
> [1] S. Dhuliawala, et. al, "A Diachronic Perspective on User Trust in AI under Uncertainty”
>
> 4. The questions are around alignment: cross-unit alignment, inter-statement alignment, and unit-to-program alignment. RaLU employs conditional generation. Though it doesn’t guarantee absolute consistency, its strength lies in dynamic context modeling and flexible dependency capture, avoiding the rigidity of hard constraints.
>
> - Corss-Unit: Units are constrained by prior ones during refinement, aided by Transformer’s self-attention for implicit semantic linking. For example, variable name changes in earlier units propagate to later ones. Hard constraints risk inconsistencies if dependencies are incomplete.
> - Code: Theoretically feasible, but RaLU prioritizes logic over syntax. Forced concatenation imposes rigid boundaries, whereas LLM regeneration dynamically optimizes interfaces (e.g., auto-completing variables) and adheres to syntax rules via pre-trained knowledge.
> - Synthesis: Conditional generation may introduce minor misalignment, but RaLU’s verification units steer the LLM toward valid solutions. Hard constraints risk overfitting to rules at the expense of semantics (e.g., redundant type conversions). Key logic (e.g., boundary handling) is preserved via attention weights, while non-critical parts are optimized—mimicking human programming cognition.
>
> 5. The decision is not made by the LLM. Each node in CFG has <= 2 children (if, else); We traverse CFG using depth-first search, organizing the nodes along the way into units to be entered in the order.

---

### Decision · Program_Chairs · 2025-05-01

**Decision:**

Accept (poster)

**Comment:**

The paper presents a novel prompting/structured reasoning approach to tackle complex reasoning tasks by aligning a set of logical units between the generated program and their corresponding NL descriptions.

Reviewers consider this idea is novel. It shows very solid improvements on several different benchmarks in the experiments.

Please update the paper for camera ready according to the review comments, e.g., provide the inference cost.